# Direct observations of $NO_x$ emissions over the San Joaquin Valley using airborne flux measurements during RECAP-CA 2021 field campaign

Qindan Zhu[1,*], Bryan Place[2,**], Eva Y. Pfannerstill[3], Sha Tong[4,5], Huanxin Zhang[4], Jun Wang[4], Clara M. Nussbaumer[1,6], Paul Wooldridge[2], Benjamin C. Schulze[7], Caleb Arata[8], Anthony Bucholtz[9], John H. Seinfeld[7], Allen H. Goldstein[3,8], and Ronald C. Cohen[1,2]

[1]Department of Earth and Planetary Sciences, University of California, Berkeley, Berkeley, CA 94720, United States
[*]Now at Department of Earth, Atmospheric and Planetary Sciences, Massachusetts Institute of Technology, Cambridge, MA, United States of America
[2]Department of Chemistry, University of California, Berkeley, Berkeley, CA 94720, United States
[**]Now at Office of Research and Development, U.S. Environmental Protection Agency, Research Triangle Park, North Carolina 27711, United States
[3]Department of Environmental Science, Policy, and Management, University of California, Berkeley, Berkeley, CA 94720, United States
[4]Department of Chemical and Biochemical Engineering, Center for Global and Regional Environmental Research, and Iowa Technology Institute, University of Iowa, Iowa City, Iowa 52242, United States
[5]Collaborative Innovation Center on Forecast and Evaluation of Meteorological Disasters (CIC-FEMD), Key Laboratory for Aerosol-Cloud-Precipitation of China Meteorological Administration, Nanjing University of Information Science Technology, Nanjing 210044, People's Republic of China
[6]Department of Atmospheric Chemistry, Max Planck Institute for Chemistry, Mainz 55128, Germany
[7]Department of Environmental Science and Engineering, California Institute of Technology, Pasadena, CA 91125, United States
[8]Department of Civil and Environmental Engineering, University of California, Berkeley, Berkeley, CA 94720, United States
[9]Department of Meteorology, Naval Postgraduate School, Monterey, CA 93943, United States

**Correspondence:** Qindan Zhu (qindan_zhu@berkeley.edu) and Ronald C. Cohen (rccohen@berkeley.edu)

**Abstract.** Nitrogen oxides ($NO_x$) are principle components of air pollution and serve as important ozone precursors. As the San Joaquin Valley (SJV) experiences some of the worst air quality in the United States, reducing $NO_x$ emissions is a pressing need, yet quantifying current emissions is complicated due to a mixture of mobile and agriculture sources. We performed airborne eddy covariance flux measurements during the Re-Evaluating the Chemistry of Air Pollutants in CAlifornia (RECAP-CA) field campaign in June 2021. Combining footprint calculations and land cover statistics, we disaggregate the observed fluxes into component fluxes characterized by three different land cover types. On average we find emissions of 0.95 mg N m$^{-2}$ h$^{-1}$ over highways, 0.43 mg N m$^{-2}$ h$^{-1}$ over urban areas and 0.30 mg N m$^{-2}$ h$^{-1}$ over croplands. The calculated $NO_x$ emissions using flux observations are utilized to evaluate anthropogenic emission inventories and soil $NO_x$ emission schemes. We show that two anthropogenic inventories for mobile sources, EMFAC (EMssion FACtor) and FIVE (Fuel-based Inventory for Vehicle Emissions), yield strong agreement with emissions derived from measured fluxes over urban regions. Three soil $NO_x$ schemes, including MEGAN v3 (Model of Emissions of Gases and Aerosols from Nature), BEIS v3.14 (Biogenic Emission Inventory System) and BDISNP (Berkeley Dalhousie Iowa Soil NO Parameterization), show substantial underestimates over the study

domain. Compared to the cultivated soil $NO_x$ emissions derived from measured fluxes, MEGAN and BEIS are lower by more than one order of magnitude and BDISNP is lower by a factor of 2.2. Despite the low bias, observed soil $NO_x$ emissions and BDISNP present a similar spatial pattern as well as temperature dependence. We conclude that soil $NO_x$ is a key feature of the $NO_x$ emissions in the SJV and that a biogeochemical process-based model of these emissions is needed to simulate emissions for modeling air quality in the region.

## 1  Introduction

Nitrogen oxides ($NO_x \equiv NO + NO_2$) are important trace gases that affect both the gas and aerosol phases of tropospheric chemistry. $NO_x$ regulates the concentrations of the primary atmospheric oxidant, hydroxyl radicals (OH), and serves as the catalyst for the formation of ozone ($O_3$). $NO_x$ also affects the formation of inorganic nitrate aerosol through the production of nitric acid ($HNO_3$) and organic nitrates ($RONO_2$) and plays a role in secondary organic aerosol (SOA) production. $NO_x$, $O_3$ and aerosol are all detrimental to human health, triggering respiratory diseases (Kampa and Castanas, 2008; Hakeem et al., 2016) and leading to premature death (Lelieveld et al., 2015).

$NO_x$ is predominantly emitted from anthropogenic sources, including light and heavy-duty transportation, fuel combustion, and biomass burning. Among these sectors transportation is the largest in the United States (EPA, 2016). Strict regulations have been implemented to control $NO_x$ emissions. Three-way catalysts have effectively reduced emissions from gasoline-powered passenger vehicles. The application of emission control systems on coal power plants has reduced $NO_x$ emissions (De Gouw et al., 2014). The California Air Resources Board (CARB) has proposed Heavy-Duty Engine and Vehicle Omnibus Regulation and Associated Amendments and target for 90% reduction in per-vehicle heavy-duty $NO_x$emission by 2031(CARB, 2016). The regulation of mobile sources leads to an increasing importance of natural $NO_x$ sources, such as lightning and soil emissions. Soil $NO_x$ is released as a byproduct of microbial nitrification and denitrification (Andreae and Schimel, 1990). While the biogeochemistry of soil $NO_x$ emission is well established, this biogenic source involves a complex interaction of soil microbial activity, soil nitrogen (N) content. Besides, agriculture activities, such as the use of fertilizers, lead to a substantial enhancement of soil $NO_x$ emissions (Phoenix et al., 2006).

Currently, the San Joaquin Valley (SJV) in California experiences some of the most severe air pollution in the United States. The SJV cities, Visalia, Fresno, and Bakersfield are among the top ten most polluted cities for both ozone and particulate matter (American Lung Association, 2020). In order to implement appropriate emission control efforts, identifying the contribution of different $NO_x$ emissions are particularly important for the SJV as it features a complex mixture of emissions from fuel combustion and soil emissions associated with agriculture. The contribution of soil $NO_x$ emissions remains highly uncertain. While Guo et al. (2020) attribute approximately 1.1% of anthropogenic $NO_x$ emissions in California to soil $NO_x$, Almaraz et al. (2018) argued that due to growing N fertilizer use, the SJV has soil $NO_x$ emissions of 24 kg of N ha$^{-1}$ year$^{-1}$, contributing 20-51% of the $NO_x$ budget of the entire state of California. Similarly, Sha et al. (2021) estimated that 40.1% of the total NOx emissions over California in July 2018 are from soils.

Airborne eddy covariance (EC) flux measurements provide a powerful tool to investigate the emission strength of atmospheric constituents at landscape scales. It has been applied to assess the surface exchanges of greenhouse houses (GHGs) including $CO_2$ and methane (CH4) (Mauder et al., 2007; Yuan et al., 2015; Sayres et al., 2017; Hannun et al., 2020). In recent years it has been extended to study emissions of volatile organic compounds and $NO_x$ over a megacity (Karl et al., 2009; Vaughan et al., 2021), vegetation (Karl et al., 2013; Misztal et al., 2014; Wolfe et al., 2015; Kaser et al., 2015; Yu et al., 2017; Gu et al., 2017), and shale gas production regions (Yuan et al., 2015). Compared to the traditional EC measurements from instruments mounted at a fixed location on a tower, wavelet-based airborne EC measurements allow for larger spatial assessment and are well suited to regions with inhomogeneous and non-stationary source distributions (Sühring et al., 2019).

In this study, we present airborne EC flux measurements obtained during seven flights of a Twin Otter aircraft over the San Joaquin Valley in California. Companion studies of $NO_x$ emissions over Los Angeles (Nussbaumer et al., 2023) and VOC (Pfannerstill et al., 2023) and GHG fluxes (Schulze et al. in prep) will be presented separately. We utilize continuous wavelet transformation to calculate the $NO_x$ flux (Sect. 3). In conjunction with footprint calculations and land classification, we explore the spatial heterogeneity of $NO_x$ emissions and identify component fluxes from the highway, urban, and soil land types (Sect. 4). We also utilize the $NO_x$ emissions derived from flux measurements to evaluate anthropogenic emission inventories and soil $NO_x$ schemes (Sect. 5).

## 2   Measurements

The airborne EC flux measurements were conducted on a Twin Otter research aircraft operated by the Naval Postgraduate School (NPS) during the Re-Evaluating the Chemistry of Air Pollutants in CAlifornia (RECAP-CA) field campaign. The RECAP-CA field campaign was conducted between June 1st to June 22nd in California, including 7 days of measurements over the San Joaquin Valley and 9 days of measurements over Los Angeles. The flight path was designed with long straight legs to ensure good quality of flux measurements (Figure S1) (Karl et al., 2013). The aircraft flew slowly, at the airspeed of 50-60 m/s, and cruised at a low height of $\sim 300$ m above ground. The aircraft took off at $\sim$11:00 local time at Burbank Airport and landed at $\sim$18:00 local time.

The standard instruments aboard the aircraft are described in (Karl et al., 2013) and include total and dew point temperature, barometric and dynamic pressures, wind direction and wind speed, total airspeed, slip- and attack angles, GPS latitude, GPS longitude, GPS altitude, pitch, roll, and heading. These measurements are at 10 Hz temporal resolution. VOCs were measured at 10 Hz time resolution by Vocus proton transfer reaction time of flight mass spectrometer (Vocus PTR-ToF-MS) as described in Pfannerstill et al. (2023). Mixing ratios of $NO_x$ were measured at 5 Hz frequency using a custom-built three-channel thermal dissociation-laser induced fluorescence (TD-LIF) instrument. The multipass LIF cells, fluorescence collection, long-pass wavelength filtering (for $\lambda >$700 nm), and photon counting details have been previously described (Thornton et al., 2000; Day et al., 2002; Wooldridge et al., 2010). Details specific to this implementation are described below.

Air was sampled from the aircraft community inlet through PFA Teflon tubing at a rate of $\sim$6 L/min and split equally between the three instrument channels. Each measured $NO_2$ by laser-induced fluorescence utilizing a compact green laser

(Spectra-Physics ExplorerOneXP 532 nm). The laser was pulsed at 80 kHz and the 1.7 Watt average power was split between the three cells. Earlier versions of the instrument used a dye laser tuned on and off a narrow rovibronic $NO_2$ resonance at 585.1 nm. Experience over a wide variety of conditions had demonstrated the off-line signal did not depend on the sample, other than from aerosol particles and that could be eliminated by adding a Teflon membrane filter. Moving to nonresonant excitation at 532 nm provided full-time coverage at 5 Hz along with lower complexity and more robust performance of the laser system. Maintaining the LIF cells at low pressure ($\sim$0.4 kPa) was no longer required to avoid line-broadening but was still desirable to extend the $NO_2$ fluorescence lifetime for time-gated photon counting to reject prompt laser scatter. Instrument zeros were run using ambient air scrubbed of $NO_x$ every 20 minutes in flight to correct for any background drift during the flights. In addition, calibrations were performed in-flight every 60 minutes using a $NO_2$ in $N_2$ calibration cylinder (Praxair, 5.5 ppm, Certified Standard grade) diluted with scrubbed air.

$NO_2$ was measured directly in the first channel, with the sample passing only through a particle filter and a flow-limiting orifice before the cell. $NO_x$ was measured in the second by adding $O_3$ (generated with 184.5 nm light and a flow of scrubbed and dried air) to convert NO to $NO_2$ before detection. A 122 cm length of 0.4 cm i.d. tubing served as the $O_3$+NO reactor, providing 4 seconds of reaction time before the orifice. The third channel was used to measure the sum of higher nitrogen oxides (e.g. organic nitrates and nitric acid) by thermal dissociation to $NO_2$ with an inline oven ($\sim$500 C) before LIF detection.

## 3   Flux and footprint calculation

### 3.1   Pre-processing

The observed 10 Hz vertical wind speeds are downscaled to 5 Hz in order to match the time resolution of $NO_x$ measurements. The full observation data set breaks into segments with continuous wind and $NO_x$ measurements. The segment window is selected if the length is larger than 10 km and the height variation is less than 200 m. We also filter out measurements when aircraft roll angles are larger than 8 degrees to avoid perturbation in the vertical wind due to aircraft activity. While most of the measurements are within the planetary boundary layer (PBL), the airplane arose above the boundary layer occasionally and these observations above PBL are removed in later analysis. The PBL heights are determined using the sharp gradient in the dew point, water concentration, toluene concentration and temperature at the soundings conducted during the voyage, and we interpolate the PBL heights to the full duration of the flight. The PBL heights agree well against the hourly PBL heights from the High-Resolution Rapid Refresh (HRRR) product (Figure. S2).

We adjust for the lag time between the meteorology measurements and the TD-LIF measurements by shifting the time of TD-LIF observation within the time window of $\pm$ 4 seconds until the covariance with the vertical wind speed is maximized (Figure. S4). As the time lag is assumed to be due to differences in the clocks of the two instruments and the transit time of air through the TD-LIF instrument, we assume that the lag time for each flight is constant. We use the median lag time from each flight for all segments collected on the same day.

## 3.2 Continuous wavelet transformation

The continuous wavelet transformation (CWT) parameterization decomposes the time series ($x(t)$) into a range of frequencies and represents it as the convolution of the time series with a wavelet function (Torrence and Compo, 1998).

$$W(a,b) = \int_{\infty}^{-\infty} x(t)\psi_{a,b}^*(t)dt \tag{1}$$

$$\psi_{a,b}^*(t) = \frac{1}{\sqrt{a}}\psi_0(\frac{t-b}{a}) \tag{2}$$

where $W(a,b)$ is the wavelet coefficient; $\psi_{a,b}^*(t)$ is the wavelet function, which is based on a "mother" wavelet $\psi_0$ and is

adjusted with a transition parameter $b$ and a scale parameter $a$. The transition parameter determines the location of the "mother" wavelet and the scale parameter defines the frequency. We use the Morlet wavelet as the "mother" wavelet.

$$\psi_0 = \pi^{-1/4}e^{6i\eta}e^{-\eta^2/2} \tag{3}$$

The Morlet wavelet has been widely applied to represent turbulence in the atmosphere due to a reasonable localization in the frequency domain and a good ability of edge detection (Schaller et al., 2017).

Time domain scales are increased linearly with the increment of the time resolution ($\delta t$, $0.2s$), and N is the number of data points. Frequency domain scales are represented by an exponential array of scale parameters $a_j$ with the increment $\delta j$ of $0.25s$. The smallest frequency scale is the Nyquist frequency, which is twice the time resolution ($0.4s$).

$$b_n = N\delta t \tag{4}$$

$$a_j = a_0 \times 2^{j\delta j} \tag{5}$$

For two simultaneous time series of $NO_x$ ($W_c(a,b)$) and vertical wind speed ($W_w(a,b)$), we first detrend them by subtracting out the average followed by dividing the standard deviation of a scalar time series. Then we obtain the wavelet cross-spectrum following Eqn. 9. The Morlet wavelet-specific reconstruction factor $C_\delta$ is 0.776. We then sum up over the full frequency scales to yield a time series of flux (Eqn. 10).

$$\bar{W}_c = \frac{1}{N}\sum_{i=1}^{N}W_{ci} \quad \text{and} \quad \bar{W}_w = \frac{1}{N}\sum_{i=1}^{N}W_{wi} \tag{6}$$

$$\hat{\sigma}_c^2 = \frac{1}{N-1}\sum_{i=1}^{N}(W_{ci}-\bar{W}_{ci})^2 \quad \text{and} \quad \hat{\sigma}_w^2 = \frac{1}{N-1}\sum_{i=1}^{N}(W_{wi}-\bar{W}_{wi})^2 \tag{7}$$

$$W_c'(a,b) = \frac{(W_c(a,b)-\bar{W}_c)}{\hat{\sigma}_c} \quad \text{and} \quad W_w'(a,b) = \frac{(W_w(a,b)-\bar{W}_w)}{\hat{\sigma}_w} \tag{8}$$

$$E_{c,w}(j) = \frac{\delta t}{C_\delta}\frac{1}{N}\sum_{n=0}^{N-1}[W_c'(a,b)\cdot W_w'^*(a,b)] \tag{9}$$

$$F(t) = \hat{\sigma}_c\hat{\sigma}_w\overline{c'w'} = \hat{\sigma}_c\hat{\sigma}_w\frac{\delta t}{C_\delta}\frac{\delta j}{N}\sum_{n=0}^{N-1}\sum_{j=0}^{J}\frac{[W_c'(a,b)\cdot W_w'^*(a,b)]}{a(j)} \tag{10}$$

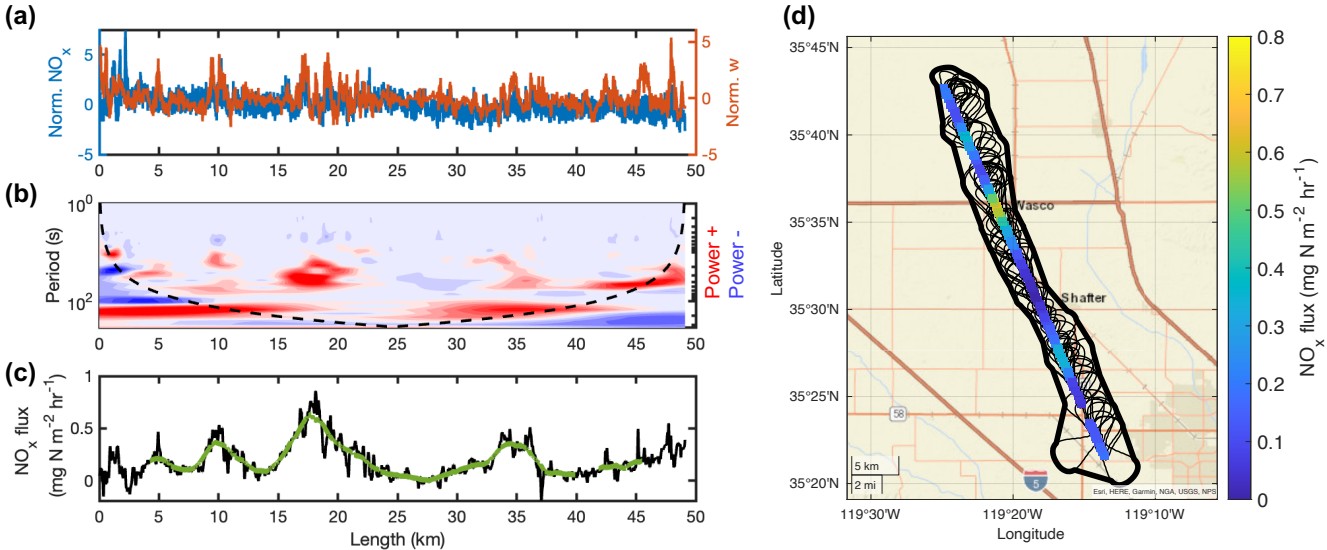

**Figure 1.** a) The variance of $NO_x$ and vertical wind speed, b) frequency and time-resolved wavelet power spectrum with the cone of influence shown as a black dotted line, c) the integrated fluxes from the raw data points are shown in black, the fluxes after moving averaging and COI filtering are shown in green. d) the distribution map of flux re-sampled at 500m. The black lines show the 90th percentiles of the footprints and the thick black line denotes the contours of all footprints.

Figure 1 exhibits an example of CWT flux calculation. Figure 1 (a) shows the detrended $NO_x$ and vertical wind speed in a straight segment of $\sim$ 50 km. The detrending is realized by subtracting out the average followed by dividing the standard deviation of a scalar time series. Both time series are decomposed using CWT algorithm to yield the cross-power spectrum shown in Figure 1 (b). Due to the finite length in time, the wavelet power spectrum is prone to higher uncertainties closer to the edge (Mauder et al., 2007). The regions of the wavelet power spectrum where the edge effects are the largest are identified as the Cone of Influence (COI). Data points containing >80% spectral power within the cone of influence are removed for quality control. The power spectrum is then integrated over all frequencies to the time series of $NO_x$ flux (Figure 1 (c)). We processed the integrated fluxes as 2 km moving averages to address the influence of large-scale turbulence and then re-sampled them at 500 m.

### 3.3 Footprint calculation

The footprint describes the contribution of surface regions to the observed airborne flux. We use the KL04-2D parameterization to calculate a space-resolved footprint map. This KL04-2D parameterization is developed from a 1-D backward Lagrangian stochastic particle dispersion model (Kljun et al., 2004). Metzger et al. (2012) implemented a Gaussian cross-wind distribution function to resolve the dispersion perpendicular to the main wind direction. The input parameters include the height of the measurements, standard deviation of horizontal and vertical wind speed, horizontal wind direction, boundary layer height,

surface roughness length, and friction velocity. We obtain the surface roughness length and friction velocity from the HRRR
product. For each flux observation, we calculate the footprint map at the spatial resolution of 500m and then extract the 90%
contour. Figure 1 (d) depicts the 90% KL04-2D footprint contours of observations resampled to 500 m in one segment. Each
footprint contour is aligned with the horizontal wind direction and is transformed into a geographic coordinate space.

### 3.4 Filter out $NO_x$ fluxes impacted by the off-road vehicle emissions

It is worth noting that croplands includes not only soil $NO_x$ emissions but the off-road vehicle emissions. Erroneously attributing the $NO_x$ from off-road vehicle emissions to soil $NO_x$ emissions leads to a high bias. While trimethylbenzene was observed during RECAP-CA field campaign, Pfannerstill et al. (2023) presented the trimethylbenzene fluxed using the same algorithm described in Sect.3.2. The trimethylbenzene fluxes are interpolated to match the $NO_x$ fluxes in time and are utilized as an indicator of off-road vehicle emissions over croplands (Tsai et al., 2014). The trimethylbeneze fluxes are categorized into two groups; the first group presents footprints covering croplands exclusively and the second group presents footprints with mixed land cover types. Shown in Figure S9, the trimethylbeneze flux is much lower over croplands, a median of 0.003 mg m$^{-2}$ h$^{-1}$ compared to a median of 0.009 mg m$^{-2}$ h$^{-1}$ over mixed land cover types including highway and urban areas. Among all observations over cropland, we identify those with the trimethylbeneze flux larger than 0.02 mg m$^{-2}$ h$^{-1}$, which consists of 7% of the total data points, are impacted by the off-road vehicle emissions, and then filter out them in the later analysis. We also vary the threshold of the trimethylbeneze flux between 0.005 mg m$^{-2}$ h$^{-1}$ and 0.04 mg m$^{-2}$ h$^{-1}$ and conclude that the choice of the threshold does not influence the results.

### 3.5 Vertical divergence

Extrapolating the airborne flux to surface flux should account for the vertical divergence. The vertical divergence is a result of multiple processes, including net in-situ production or loss, storage, and horizontal advection.

To investigate the impact of vertical divergence, the flight route includes three vertically stacked racetracks, during which the segments are close to each other in space but vary in height. After removing the legs that fail the quality control, only one racetrack measurement carried out between 14:20 to 15:10 on June 8th presented qualified flux segments, and the vertical distribution of fluxes is shown in Fig. S4. No consistent increase or decrease of fluxes with increasing height is detected during the racetrack in this study because the vertical divergence is hampered by emission heterogeneity. Shown in Fig. S7, the footprint map for each segment at various altitudes covers regions with high heterogeneity. Therefore, we use an alternative approach to calculate the vertical divergence. Instead of extracting racetrack measurements, we collect a subset of flux measurements during the whole field campaign based on the footprint coverage. Only fluxes with footprints covering croplands exclusively are included to avoid emission heterogeneity. We calculate the ratio of measurement heights relative to the PBL height ($z/z_i$) and 98% of selected fluxes are located within 70% of the PBL height and they are divided into 7 bins of $z/z_i$ with uniform width. We then perform a linear fit for the binned median fluxes versus $z/z_i$ to calculate the vertical correction factor ($C = \frac{slope}{intercept}$). This correction factor is used to linearly extrapolated the fluxes at the measurement height ($F_z$) to fluxes at the surface ($F_0$)

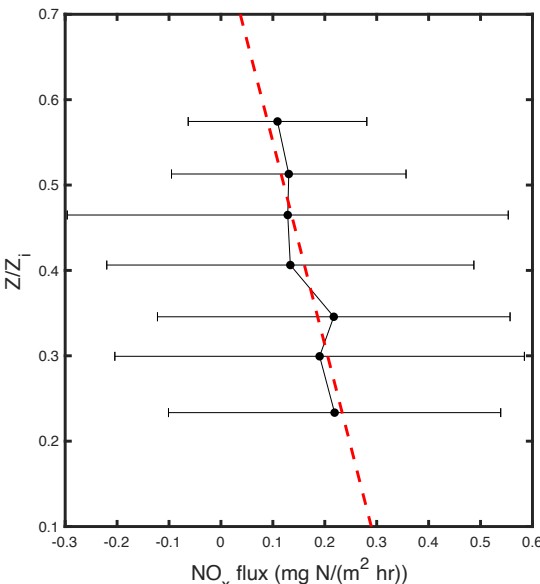

**Figure 2.** Vertical profiles of measured fluxes above croplands during RECAP-CA field campaign binned by the ratio of measurement height and PBL height ($z/z_i$). The points represent the median flux within each bin, and the error bars represent the standard deviation. The red dashed line shows a linear fit for median fluxes versus relative height.

(Eqn. 11). After vertical divergence correction, the surface fluxes are on average 26% higher than the fluxes at the measurement heights.

$$F_0 = \frac{F_x}{1 + C\frac{z}{z_i}} \tag{11}$$

### 3.6   Data qualify control and uncertainty analysis

The flux detection limit does not only depend on the signal-to-noise ratio of the $NO_x$ measurement, but also varies with wind speed and atmospheric stability. Following Langford et al. (2015), we calculate the detection limit of flux (LoD) before the moving and spatial average are applied. For each segment, the observed $NO_x$ is replaced with a white noise time series and is then feed into the CWT to yield the corresponding time series of "noise" flux. The random error affecting the flux ($\sigma_{NO_x,noise}$) is defined as the standard deviation of this noise-derived flux, and LoD is defined as $2\times\sigma_{NO_x,noise}$ ($95^{th}$ confidence level).

Among 142 segments, Figure 3 (a) shows the distribution of flux LoD among 142 segments. The LoDs range from 0.02 mg N $m^{-2}$ $h^{-1}$ to 0.30 mg N $m^{-2}$ $h^{-1}$, and the average LoD is 0.10 mg N $m^{-2}$ $h^{-1}$. To obtain a better constraint on the flux quality, we compare the LoD against the time series of flux in each segment and filter out 18 segments in which the whole time series is below the LoD.

The flux calculation using CWT introduces uncertainty from a variety of sources. We describe systematic errors and random errors following Wolfe et al. (2018). Systematic errors arise from the under-sampling of high-frequency and low-frequency ranges. The CWT algorithm fails to resolve a frequency higher than the Nyquist frequency. Due to the high temporal resolution (5 Hz), we expect a minimal loss at the high-frequency limit (Figure. S5). The upper limit of systematic error associated with low frequency is calculated using Eqn. 12 (Lenschow et al., 1994).

$$SE \leq 2.2(\frac{z}{z_i})^{0.5}\frac{z_i}{L} \tag{12}$$

$z$ and $L$ are the measurement heights and the length of segments, respectively. $z_i$ are the boundary layer heights from HRRR. We calculate the low-frequency error ranges from 1%-5%.

Random errors arise from the noise in the instrument ($RE_{noise}$) as well as the noise in turbulence sampling ($RE_{turb}$), which are calculated using Eqn. 13 and Eqn. 14 (Wolfe et al., 2018; Lenschow et al., 1994).

$$RE_{noise} = \frac{\sqrt{\sigma_{NO_x,noise}^2 \sigma_w^2}}{N} \tag{13}$$

$$\frac{RE_{turb}}{F} \leq 1.75(\frac{z}{z_i})^{0.25}(\frac{z_i}{L})^{0.5} \tag{14}$$

$z$, $L$ and $z_i$ are the same as Eqn. 12, $\sigma_w^2$ is the variance of vertical wind speed. Note that $RE_{noise}$ assumes the noise in each time step is uncorrelated, therefore, we ignore the moving average step in the uncertainty calculation and $N$ denotes the number of points used to yield each 500m spatially averaged flux.

Utilizing a constant lag time introduces an additional source of uncertainty. We estimate the uncertainty by comparing the calculated fluxes using segment-specific and constant lag times across all segments that specific lag times are available. Shown in Figure. S4, the difference is less than 25% for 90 percent of the data. Therefore, we attribute an uncertainty of 25% due to the lag time correction ($RE_{lag}$). While we believe this error is unphysical and that a single lag time is more appropriate, we include it to be conservative in our estimate of the uncertainties.

Estimating the uncertainty caused by the correction of vertical divergence is tricky. While we conclude that the influence of vertical divergence is non-negligible, it is ignored in some previous airborne flux studies (e.g. Vaughan et al., 2016; Hannun et al., 2020; Vaughan et al., 2021; Drysdale et al., 2022). While the flux is scattered in each vertical intervals in our divergence calculation, we first bootstrap the flux observations and calculate the uncertainty of correction factor ($\sigma_C$) to 40%. As we see a significant difference in vertical correction factor on racetrack measurements versus a selected subset of flux observations, we tentatively set the uncertainty of $C$ to 100%, in order to account for the case of no vertical divergence. Besides, we account for a 30% uncertainty in the PBL heights.

We propagate the total uncertainty from each component using Eqn. 16 and the distribution of total uncertainty is shown in Figure 3 (b). The average uncertainty is 60% and the interquartile of total uncertainty are 48% and 68%. The random error and the vertical divergence correction dominate the uncertainty and the uncertainty is consistent with previous studies (Wolfe et al., 2018; Vaughan et al., 2016).

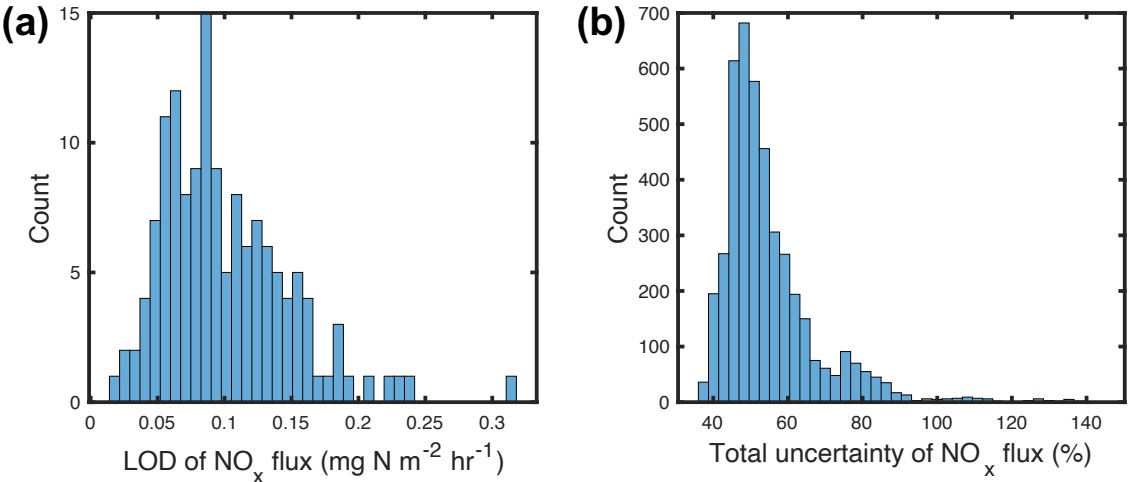

**Figure 3.** a) The distribution of segment-based $NO_x$ flux detection limit (LoD). b) The distribution of total uncertainty of $NO_x$ flux.

$$\sigma_{F_z} = \sqrt{SE^2 + RE_{noise}^2 + RE_{turb}^2 + RE_{lag}^2} \tag{15}$$

$$\sigma_{F_0} = \sqrt{\frac{\sigma_{F_z}^2}{(1+C\frac{z}{z_i})^2} + \sigma_C^2(\frac{z}{z_i})^2(\frac{F_z}{(1+C\frac{z}{z_i})^2})^2 + \sigma_{z_i}^2(\frac{Cz}{z_i^2})^2(\frac{F_z}{(1+C\frac{z}{z_i})^2})^2} \tag{16}$$

## 4 Component flux disaggregation

The overview of observed fluxes across 7 flights over San Joaquin Valley is illustrated in Fig. 4. It shows a distinct spatial heterogeneity (Figure 4 (a)). For instance, high $NO_x$ flux signals are detected when the aircraft was flying above highway 99 between Bakersfield and Visalia. The transect of cities, such as Fresno, capture a substantial enhancement of $NO_x$ fluxes. Figure 4 (b) exhibits the distribution of airborne fluxes. 90% of the fluxes are positive, demonstrating that our airborne flux measurements are capable of detecting $NO_x$ emissions over the study domain. We attribute the remaining 10% of negative fluxes to the uncertainties in the flux calculation. The distribution of observed fluxes is right-skewed; the mean and median observed flux over the SJV is 0.37 mg N m$^{-2}$ h$^{-1}$ and 0.25 mg N m$^{-2}$ h$^{-1}$, respectively. The interquantile range of flux is 0.11 mg N m$^{-2}$ h$^{-1}$ and 0.49 mg N m$^{-2}$ h$^{-1}$. 1.2% of extremely high fluxes exceeding 2 mg N m$^{-2}$ h$^{-1}$ represents the long tail in the flux distribution, which are, like the negative fluxes, most likely caused by the incomplete sampling of the spectrum of eddies driving the fluxes.

As discussed in Sect. 3.3, we then calculate the footprint for each flux observation during the RECAP field campaign. Figure 4 (a) shows the 90% footprint extent in grey. Fig. S8 shows that the 90% extent for the calculated footprints ranges from 0.16 to 12 km with a mean extent of 2.8 km. The KL04-2D footprint algorithm has been applied to airborne flux analysis over

London and in that study, the 90% footprint extents range from 3 km to 12 km from the measurement (Vaughan et al., 2021). While the largest footprint extent is comparable with those from Vaughan et al. (2021), our calculated footprints mostly have a smaller extent as 62% of the footprint extents are within 3 km of the aircraft flight track. We attribute the small footprints to the stagnant weather conditions and weaker horizontal wind advection compared to London. The mean wind speed is 2.9 m/s for full observation data sets and 2.4 m/s for those data points with footprint extents less than 3 km. The largest footprint extent corresponds to observations at the foothills, due to higher altitude relative to the boundary layer height and stronger horizontal wind advection.

The region covered by the footprints is composed of mixed land cover types. We use the 2018 USDA CropScape database (https://nassgeodata.gmu.edu/CropScape/) to describe the land cover types. The resolution has been degraded from the native 30m resolution to 500m. For each grid, the land cover type is assigned if a land type makes up more than 50% of the 500m grid cell. We generalize a "soil" land cover type if the land cover type is identified as either cropland or grassland. The grids classified as "developed" in CropScape are dominated by anthropogenic activities including transportation and fuel combustion. We overlay the national highway network and categorize the grids containing highways as "highway" land types. The remaining grids are classified as "urban" and they correspond to the area with heavy populations in the absence of highways. The distinction between "highway" and "urban" land type is utilized to address on-road mobile sources. 37% of the flux observations include the highway land type in the 90% footprint extent, 23% of the observations include the urban land type and 96% of the observations include cultivated soil land type.

To disentangle the flux emanating from different land cover types, we apply the Disaggregation combining Footprint analysis and Multivariate Regression (DFMR) methodology described in Hutjes et al. (2010). The observed fluxes are treated as the weighted sum of component fluxes from each land cover type:

$$F_{obs} = \sum_{k=1}^{3} w_k F_k \qquad (17)$$

where $k_1$ to $k_3$ denote highway, urban, and soil land types, $w_k$ is the fractional area within the 90% footprint contour and $F_k$ are the corresponding component fluxes. The multi-linear regression is applied to observations from all flights, consisting of 4391 data points. We perform the Monte Carlo simulation to identify the uncertainty of the multi-linear regression due to the flux uncertainty. The resulting statistical uncertainty is shown in Fig.5. The highway land type yields the highest flux of 0.96 mg N m$^{-2}$ h$^{-1}$ with a standard deviation of 0.04 mg N m$^{-2}$ h$^{-1}$. The areas classified as urban land type exhibit a flux of 0.43 ($\pm$0.02) mg N m$^{-2}$ h$^{-1}$, which is $\sim$ 50% of the highway flux. Most likely the fluxes from highway are even higher than 0.96 mg N m$^{-2}$ h$^{-1}$. Note that the land type map is at 500m spatial scale, the grid classified as highway indeed includes both highway and areas near the highway. If, for example, the highway is only 10% of the true area of the land cover pixel, then the fluxes on the highway could be as much as 10 times larger. The cultivated soil land type flux of 0.30 ($\pm$0.01) mg N m$^{-2}$ h$^{-1}$ is large. It is about 1/4 the magnitude of the highway flux and half that of the urban flux. As the total area of soil pixels are much larger than the area of highway or urban pixels, integrated across the SJV, cultivated soil NO$_x$ emissions are a major factor.

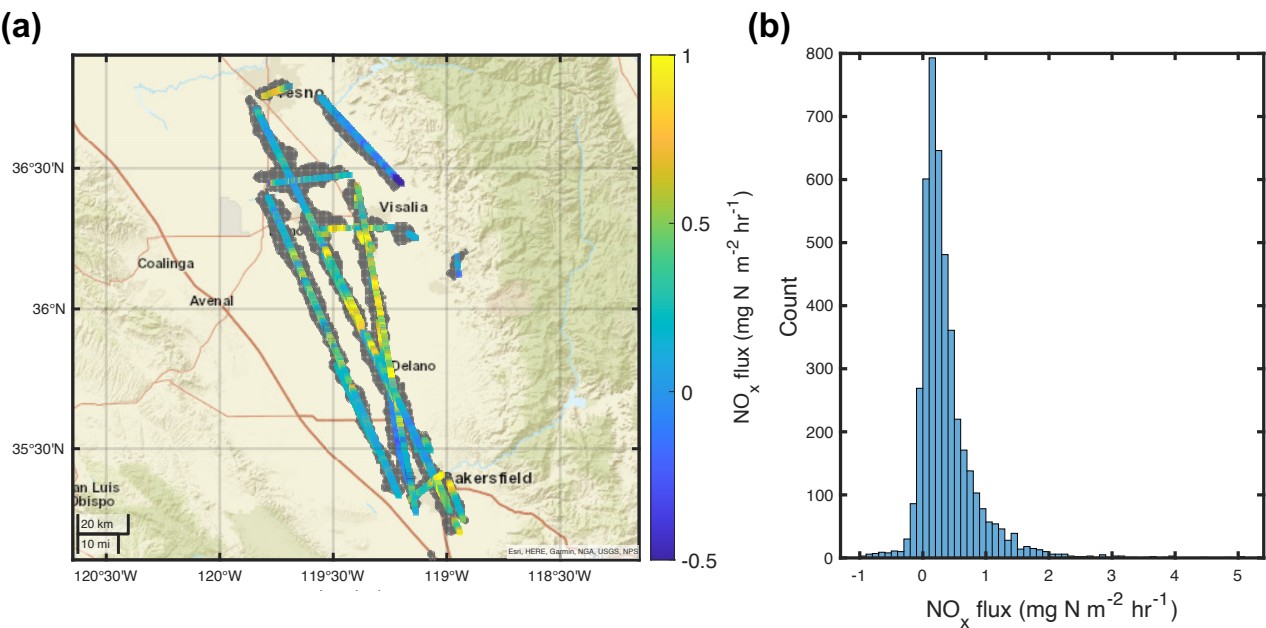

**Figure 4.** a) The map of observed airborne fluxes over 7 flights over the San Joaquin Valley. If the segment overlaps each other, the average flux is calculated. The grey shade represents the coverage of 90% footprint extents for all flux observations. b) The distribution of full data sets of observed airborne $NO_x$ fluxes.

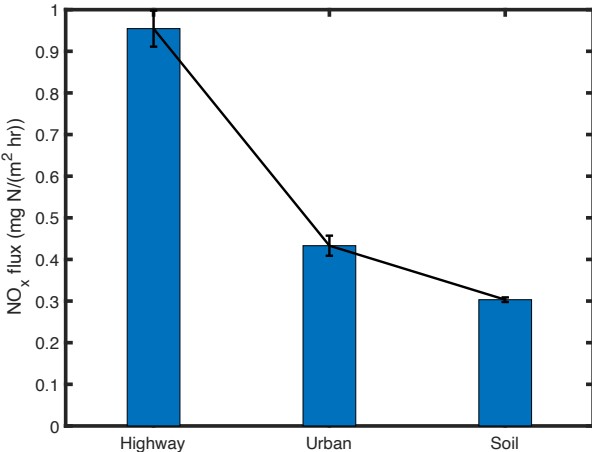

**Figure 5.** Bootstrapped statistical results of multi-linear regression to resolve component fluxes from the highway, urban, and cultivated soil land types. Each bar represents the average component fluxes from each land type and the black line shows the standard deviation.

## 5 Calculation of $NO_x$ emission map using airborne $NO_x$ fluxes

While these separate component fluxes emphasize the distinction between individual land types at the spatial resolution of the land cover (500m), we utilize the $NO_x$ fluxes to yield an estimate of $NO_x$ emission at 4km. For each 4 km grid, we collect the observed fluxes whose 90% of the footprint overlaps with this grid area and define the weight $r_k$ as the fractional area that the footprint covers. The emission, in a unit of mg N m$^{-2}$ h$^{-1}$, is calculated by the weighted average of flux (Eqn. 18). Only grids measured by at least five flux observations are considered in order to focus our attention on those pixels for which we have a statistically representative sample of the emissions.

$$Emis_i = \frac{\sum_{k=1}^{n \geq 5} r_k F_k}{\sum_{k=1}^{n \geq 5} r_k} \tag{18}$$

The emission is calculated based on the observations from six flights during weekdays(Fig. 6 (a)). The largest reported weekday emission was on June 03 when the median emission was 0.39 mg N m$^{-2}$ h$^{-1}$. The lowest weekday emission was observed on June 15 with the median emission of 0.14 mg N m$^{-2}$ h$^{-1}$. The large daily variation observed in estimated emissions during weekdays is partially due to the variation in flight routes and footprint coverage. This is illustrated by the daily estimated emission map shown in Fig. S10.

As the emission inventories make a distinction between weekdays and weekends and do not account for the daily variation on different weekdays, we average over the six-weekday flights to yield the best estimate of emission maps over the San Joaquin Valley derived from flux measurements (Fig. 6 (b)). The median estimated weekday $NO_x$ emission over the study domain is 0.26 mg N m$^{-2}$ h$^{-1}$ with the interquantile range of 0.14 and 0.46 mg N m$^{-2}$ h$^{-1}$. The observed emission map describes high $NO_x$ emissions in the cities of Bakersfield (119°W, 35.3°N) and Fresno (119.8°W, 36.75°N) and along highway 99.

### 5.1 Evaluation of anthropogenic $NO_x$ emission inventories

First, we compare the observations to the inventory developed by the California Air Resources Board (CARB). The anthropogenic emissions of $NO_x$ consist of mobile sources, stationary sources, and other $NO_x$ emissions from miscellaneous processes such as residential fuel combustion and managed disposal. In the CARB inventory, the mobile sources are estimated from EMission FACtor (EMFAC) v1.0.2 (CARB, 2021a) and OFFROAD mobile source emission models (CARB, 2021b). The stationary sources are estimated based on the reported survey of facilities within each local jurisdiction and the emission factors from California Air Toxics Emission Factor (CATEF) database (CARB, 2021c). Hereinafter we utilize "EMFAC" to represent anthropogenic vehicle-related $NO_x$ emissions used in the CARB inventory. An alternative anthropogenic emission inventory is the fuel-based inventory for vehicle emissions (FIVE), developed by McDonald et al. (2012) and updated by Harkins et al. (2021). Both emission inventories are at 4 km spatial resolution.

To disentangle the contribution of different $NO_x$ emissions sources, we attribute emissions at grid cells covering either highway or urban regions to anthropogenic emissions from transportation and fuel combustion, and those at remaining grid cells are categorized as soil $NO_x$ emissions. For each grid cell categorized as anthropogenic emission dominant, we then match the emission inventories representing the weekday scenario to the same hour and grids of emissions derived from measured

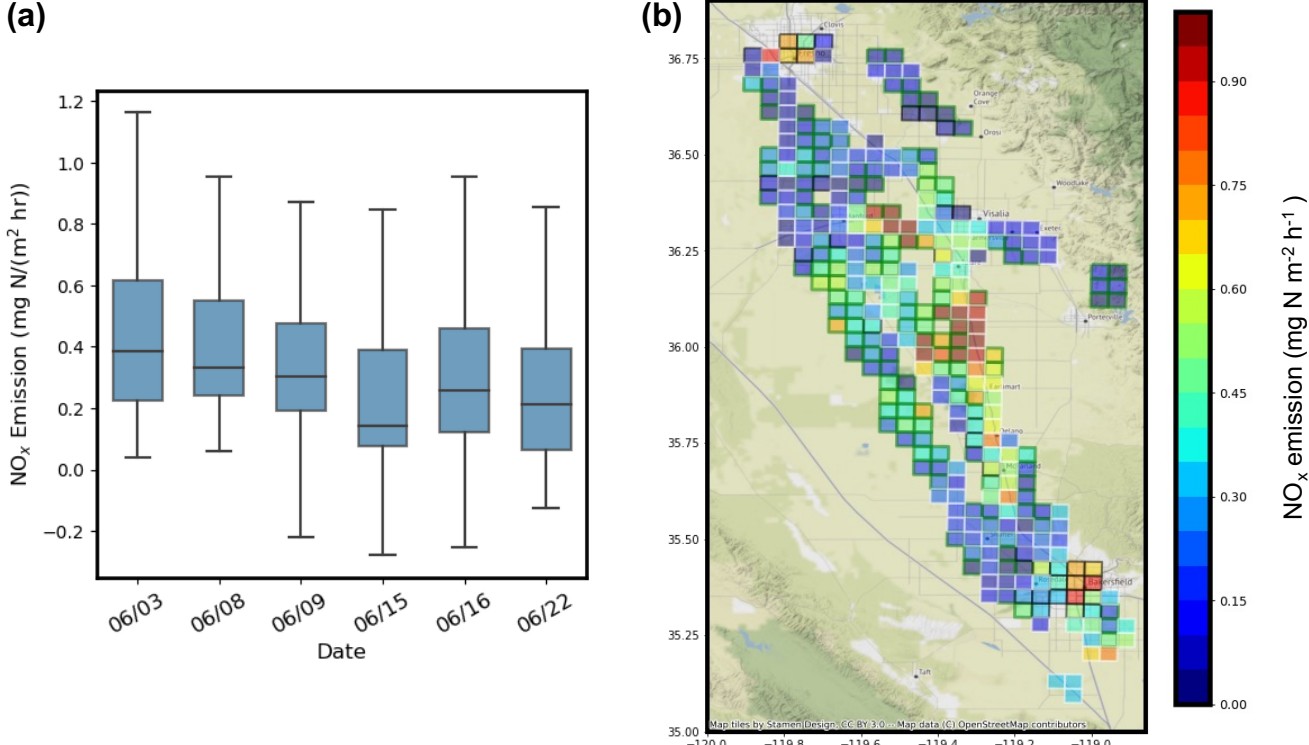

**Figure 6.** a) The whisker box plot of observed emissions for each flight, aligned in the order of flight days. The box represents the interquartile ranges of observed emissions and the line represents the median emission. The whiskers show the maximum and minimum values. b) The spatial distribution of emission at 4 km over SJV derived from observed fluxes during weekdays. The patch color shows the observed $NO_x$ emission. The edge color denotes the land cover type; the grid cells covering highways in white, those covering urban regions in black, and the rest of the grid cells that are categorized with cultivated soil land cover types in green. ©OpenStreetMap contributors 2022. Distributed under the Open Data Commons Open Database License (ODbL) v1.0.

fluxes. The corresponding hour of this estimated emission is rounded to the closest hour of the observation times. Figure 7 and Figure S12 show the comparison of observed anthropogenic emissions against EMFAC and FIVE emission inventories. Over urban regions, the mean and median observed RECAP $NO_x$ emission are 0.37 mg N m$^{-2}$ h$^{-1}$ and the interquartile range is 0.14 and 0.58 mg N m$^{-2}$ h$^{-1}$. Both EMFAC and FIVE yield a good agreement with our measurements; the mean urban $NO_x$ emission are 0.40 and 0.43 mg N m$^{-2}$ h$^{-1}$. However, the median urban $NO_x$ emission in these inventories is 24% and 22%

lower than the observation, respectively. The estimated $NO_x$ emission on grid cells covering highways is more scattered. The median estimated $NO_x$ emission is 0.24 mg N m$^{-2}$ h$^{-1}$. It is lower than on urban grid cells due to spatial averaging and the fact that most of the highway length is outside the urban regions. The distribution of observed RECAP $NO_x$ emissions from the highway is right-skewed, characterized by an interquartile range of 0.14 and 0.47 mg N m$^{-2}$ h$^{-1}$. We also note that over

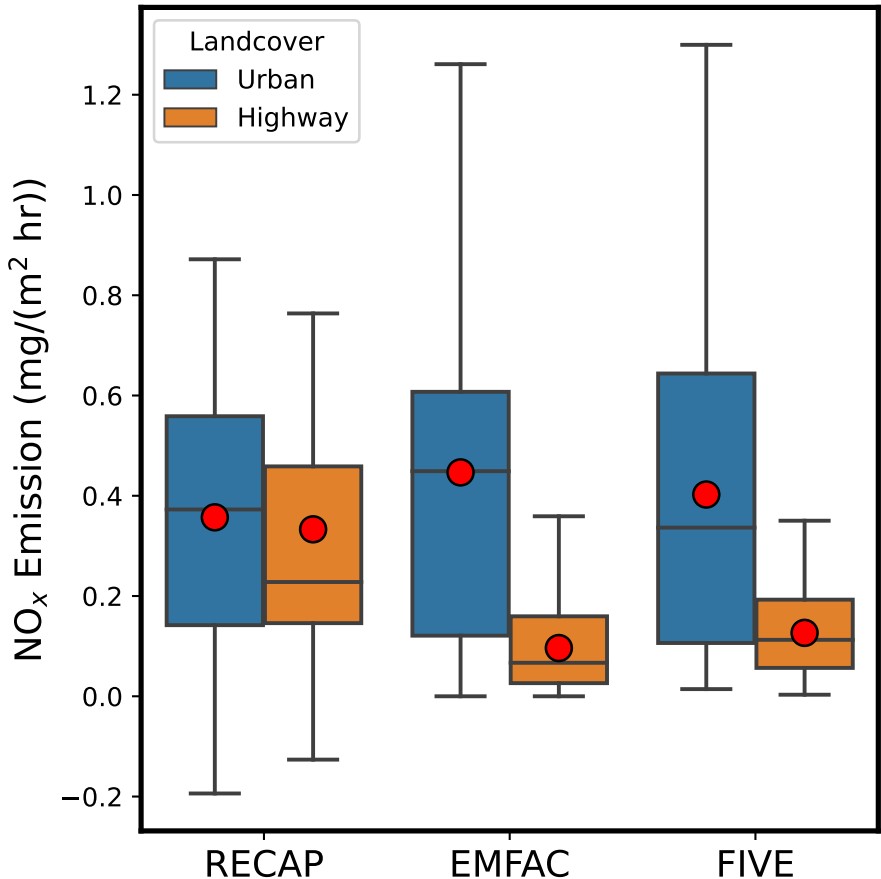

**Figure 7.** Whisker box plot of observed RECAP anthropogenic $NO_x$ emissions from transportation and fuel combustion as well as those from EMFAC and FIVE emission inventories, separated by highway and urban land cover types. The box is the interquartile range with the line of the median value. The maximum and minimum emissions are shown by whiskers and the mean emissions are shown in red dots.

Highway 99, the RECAP $NO_x$ emission is a factor of 3 higher than average on grid cells near congestion, reflecting the variation of emission caused by real-time traffic conditions. Both EMFAC and FIVE provide lower $NO_x$ emissions over highway grids, the median $NO_x$ emissions are 37% and 50% of those from the RECAP observations. The highway pixels include a land cover that is mostly non-highway; typically soil. If soil N emissions are substantially larger than in these inventories, it is possible that the measurements and bottom-up inventories for highways are in better agreement than indicated by the figure.

## 5.2 Evaluation of soil $NO_x$ scheme

Soil $NO_x$ emissions are determined by biogeochemical processes including soil microbe-mediated nitrification and denitrification. Process-based biogeochemical models have been developed to mechanistically represent soil $NO_x$ emissions by sim-

ulating nitrogen interactions in ecological systems, such as DeNitrification-DeComposition (DNDC) (Li et al., 1992, 1994; Guo et al., 2020) and DayCENT (Del Grosso et al., 2000; Rasool et al., 2019). However, these process-level models are not yet widely applied to chemical transport models, and the default model configuration uses empirical soil $NO_x$ schemes. The Model

of Emissions of Gases and Aerosols from Nature v3 (MEGAN) (Guenther et al., 2012) is the most commonly used scheme and is used to predict soil $NO_x$ emissions in the CARB emission inventory. It is gridded at 4 km spatial scale and has hourly time steps. The Biogenic Emission Inventory System (BEIS) is the default scheme to estimate volatile organic compounds from vegetation and NO from soil developed by the United States Environmental Protection Agency (EPA). We obtain the hourly BEIS v2.14 soil $NO_x$ emission at 4 km during the study period from the Weather Research and Forecasting-Chemistry

model (WRF-Chem) with the same model configuration described in Kim et al. (2022). While soil $NO_x$ varies nonlinearly with meteorological conditions, soil conditions, and agricultural activities, both MEGAN and BEIS simplify the nonlinearity to an activity factor ($\gamma$), as a function of ambient temperature, leaf area index, and leaf age. A recently developed soil $NO_x$ scheme, the Berkeley Dalhousie Iowa Soil NO Parameterization (BDISNP) (Hudman et al., 2012; Sha et al., 2021), is an intermediate complexity model and includes more details than the other two in order to be more faithful to direct measurements made at

soils and to describe their seasonal and hourly variations. The BDISNP includes parameters representing the effects of soil moisture, temperature, and soil nitrogen including fertilizer. Using the same WRF-Chem setup described in Sha et al. (2021), we also calculate the BDISNP soil $NO_x$ emissions during the study period at the spatial resolution of 2 km and re-grid them to 4 km.

Figure. 8 (a) illustrates the range of soil N emissions derived from RECAP observations as compared to these three different

soil $NO_x$ schemes. The analysis of the observations exhibits a median cultivated soil $NO_x$ emission of 0.26 mg N m$^{-2}$ h$^{-1}$; the interquartile range of the inferred emission is 0.14 mg N m$^{-2}$ h$^{-1}$ and 0.45 mg N m$^{-2}$ h$^{-1}$. MEGAN and BEIS both have an order of magnitude lower emissions with median soil $NO_x$ emissions of 0.008, 0.011 mg N m$^{-2}$ h$^{-1}$, respectively. The BDISNP soil $NO_x$ scheme shows a median soil $NO_x$ emission of 0.14 mg N m$^{-2}$ h$^{-1}$. Figure. 8 (b) exhibits a point-by-point comparison of the observed RECAP and the BDISNP soil $NO_x$ emissions showing that there is a correspondence between the

two but the model is 2.2 times lower than the observations. Figure S13 (a) and (d) shows the spatial distribution of soil $NO_x$ emissions from observation and BDISNP scheme. Both show higher soil $NO_x$ emissions between 35.75 °N and 36.25 °N.

A distinct characteristic of soil $NO_x$ emission is its temperature dependence. For instance, Oikawa et al. (2015) identified unusually high soil $NO_x$ emissions in a high-temperature agricultural region based on in-situ observations. The temperature-driven increase in soil $NO_x$ emission raises concerns in the future warmer climate, resulting in a larger contribution to $O_3$

pollution (Romer et al., 2018). Here we leverage our flux observations to probe this temperature dependence. We collect observed $NO_x$ emissions for each flight and select the subset of $NO_x$ emissions on grids categorized as cultivated soil land type. We also collect corresponding mean soil temperature from WRF-Chem and match them to observed $NO_x$ emissions both in time and space. A range of soil temperature between 295K to 304K is observed. We then bin observed soil $NO_x$ emissions to three soil temperature categories, each of which has 4K intervals. The median soil $NO_x$ emissions increase from 0.22 mg

N m$^{-2}$ h$^{-1}$ to 0.29 mg N m$^{-2}$ h$^{-1}$ with the median soil temperature increasing from 296 K to 300 K. As the response to soil temperature is incorporated in the BDISNP scheme, we also bin the BDISNP parameterized soil $NO_x$ emissions into the

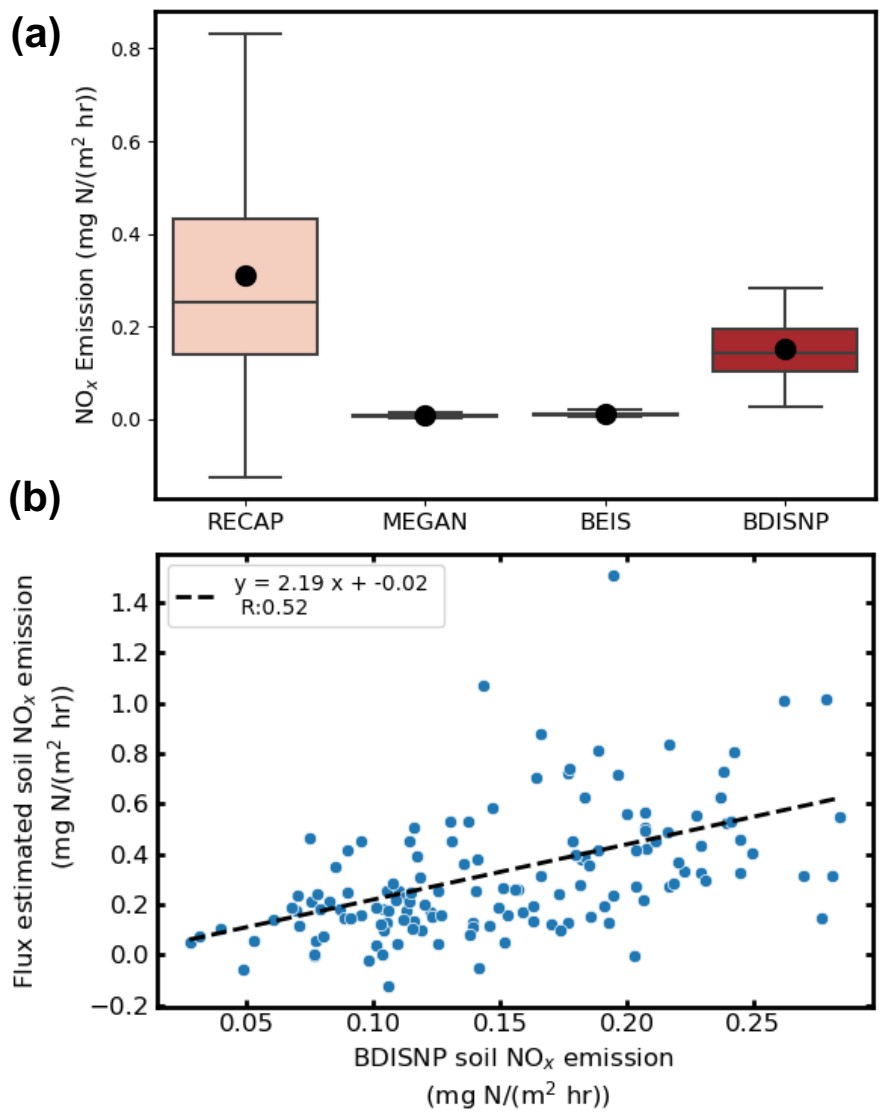

**Figure 8.** a) The whisker box plot of observed soil $NO_x$ emissions and parameterized soil $NO_x$ emissions from MEGAN, BEIS and BDISNP schemes. The mean soil $NO_x$ emissions are shown in black dots. b) The scatter plot of soil $NO_x$ emissions calculated from BDISNP scheme and from flux measurements. The dashed black is the least-square linear fit.

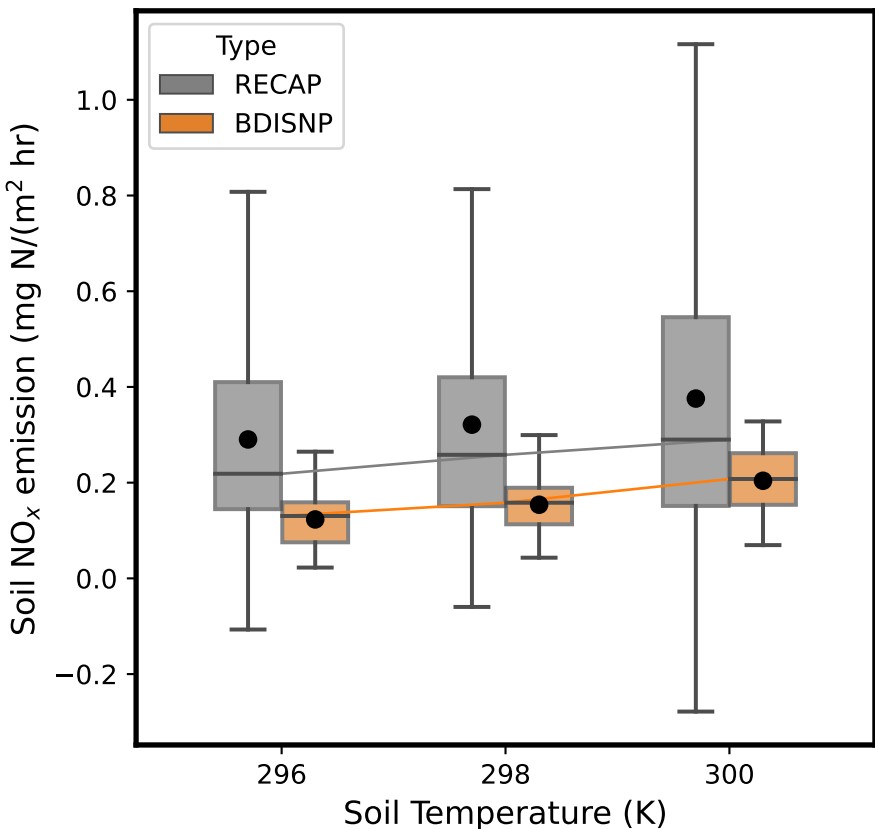

**Figure 9.** The dependence of soil $NO_x$ emissions on soil temperature from both flux measurements (gray) and BDISNP scheme (orange). Both observed and BDISNP soil $NO_x$ emissions are binned based on mean soil temperature from WRF-Chem. Three soil temperature bins are described with 4k intervals. The whisker box shows the distribution and the black dot shows the mean within each bin, the line connects median soil $NO_x$ emissions across three bins.

same soil temperature categories. Both the RECAP measured flux and the BDISNP modeled soil $NO_x$ emissions exhibit an approximately 33% increase over the range of soil temperature shown.

### 5.3 Discussion of soil $NO_x$ emissions

Soil $NO_x$ emissions in California have been studied in field experiments. Matson et al. (1997) measured soil $NO_x$ emissions from nine dominant crop types in SJV and reported mean fluxes of 0.01-0.09 mg N m$^{-2}$ h$^{-1}$. They also reported a large variation of measured $NO_x$ flux among crops and among different fields of the same crop; the highest measured $NO_x$ flux is 0.17 mg N m$^{-2}$ h$^{-1}$ due to the fertilizer application and soil moisture characteristics. Horwath and Burger (2013) observed an average flux of 0.05-0.28 mg N m$^{-2}$ h$^{-1}$ at mid-days during summertime from five crops in California, and the highest $NO_x$

flux is >4 mg N m$^{-2}$ h$^{-1}$ in systems receiving large N inputs resulting in high concentrations of ammonium. Oikawa et al.

(2015) observed soil $NO_x$ emissions in a high-temperature fertilized agricultural region of the Imperial Valley, CA, ranging between -0.02 and 3.2 mg N m$^{-2}$ h$^{-1}$. They also conducted control experiments to investigate the soil $NO_x$ emission responses to fertilization and irrigation. The highest soil $NO_x$ flux was reported ~10 days after the fertilizer at the soil volumetric water content of 30% and the soil temperature of ~313K.

The mean soil $NO_x$ flux, 0.32 mg N m$^{-2}$ h$^{-1}$, derived in our flux measurements is higher than the mean fluxes reported in Matson et al. (1997) and Horwath and Burger (2013), however, the range of estimated soil $NO_x$ flux is within those in Horwath and Burger (2013) and Oikawa et al. (2015). Fertilizer is likely the primary contributor to the higher mean soil $NO_x$ flux in our study. The RECAP-CA field campaign was conducted in June, right after the month of peak fertilizer use in SJV (Guo et al., 2020). Shown in Oikawa et al. (2015), soil $NO_x$ flux can increase up to 5-fold within 20 days of fertilizer. The higher mean soil
$NO_x$ flux is also contributed by higher soil temperature. In our study, the mean soil temperature is 299K with a range between 295K and 304K, whereas the observations in Horwath and Burger (2013) and Oikawa et al. (2015) spread over a wider range of soil temperature, 288K-315K. Consistent with our study, the temperature dependence of soil $NO_X$ emission is observed in these field experiments. Horwath and Burger (2013) reported a 2.5-3.5 fold increase in $NO_x$ fluxes with 10-degree increase in soil temperature. Oikawa et al. (2015) showed that the temperature dependence of soil $NO_x$ emission is non-linear; a steeper
increase in soil $NO_x$ emission was observed with the soil temperature exceeding 295K.

It is worth noting the limitation of estimated soil $NO_x$ emissions in our study. First of all, we are unable to investigate the dependence of soil $NO_x$ emissions on meteorological drivers other than soil temperature, such as soil moisture, as modeled soil moisture presents very small variation during the field campaign. Second, as our measurements only cover limited cropland areas in SJV over a short time period and it is around the time of fertilizer use, we cannot scale the estimated soil $NO_x$ emission
to the whole year or to the total cropland areas in California. Last, in the absence of ozone and PM2.5 observations, we cannot investigate the impact of soil $NO_x$ emission on air quality. However, as the SJV is in the $NO_x$ limited regime (Pusede et al., 2014), we expect a model that captures the soil $NO_x$ more accurately will produce higher ozone. Future work is needed to further advance our understanding of soil $NO_x$ emission and its role in urban and rural air pollution.

## 6    Conclusions

We performed airborne $NO_x$ flux measurements during RECAP-CA field campaign over the San Joaquin Valley. Seven flights were made over the SJV in June 2021. When combined with footprint and land cover information, we resolve spatial heterogeneity in landscape flux. The component fluxes are estimated based on the multi-linear regression and exhibit statistically significant differences. The component fluxes are the highest from highways at 0.96 mg N m$^{-2}$ h$^{-1}$. Cultivated soil land types emit a non-negligible flux of 0.30 mg N m$^{-2}$ h$^{-1}$. The airborne flux observations are projected to a 4 km grid spacing to yield
an estimated emission map over the SJV. We utilize this map to evaluate emission inventories commonly used in photochemical modeling. The anthropogenic emission inventories, EMFAC and FIVE, agree well with estimated mean $NO_x$ emissions over urban regions. However, the widely used, but not biogeochemical process-based, models for soil $NO_x$ emissions underestimate emissions by an order of magnitude or more in the SJV, leading to a poor assessment of the relative roles of mobile and

agriculture sources of $NO_x$ in the region. The BDISNP model as adapted by Sha et al. (2021) results in a better comparison with the observations. Even though it is still lower by a factor of 2, we show it yields a similar spatial pattern and soil temperature dependence as observed. Variations of this model are embedded in CMAQ (Rasool et al., 2019) and GEOS-CHEM (Wang et al., 2021) and have been implemented in WRF-CHEM by(Sha et al., 2021). Studies, where soil $NO_x$ is potentially important, should make use of these codes, all of which are more consistent with observations at multiple scales.

*Code and data availability.* The measurement data from the RECAP field champaign is available at https://csl.noaa.gov/projects/sunvex/. The analysis codes for this study are available at https://github.com/qdzhu/FLUX/.

*Author contributions.* RCC and AHG supervised the research; BP, EP, BS, PW CA, AB, JS, RCC, AHG participated in the field campaign; BP and PW conducted the $NO_x$ measurements; ST, HZ and JW provided model simulated BDISNP soil $NO_x$ emissions; QZ performed the analysis with contributions from BP, EP, CN; QZ prepared the manuscript; all authors have reviewed and edited the paper.

*Competing interests.* The authors have the following competing interests: At least one of the (co-)authors is a member of the editorial board of Atmospheric Chemistry and Physics.

*Acknowledgements.* This RECAP field campaign was funded by California Air Resources Board Contract number 20RD003, 20AQP012, and Presidential Early Career Award for Scientists and Engineers (PECASE) (from Brian McDonald). QZ was supported by NOAA Climate & Global Change Postdoc Fellowship and EP was supported by a Feodor Lynen Fellowship of the Alexander von Humboldt Foundation. JW and HZ acknowledge the support of NASA Atmospheric Composition and Modeling Program (ACMAP, grant : 80NSSC19K0950) We thank Dennis Baldocchi, Glenn Wolfe, Erin Delaria for their help in calculating vertical divergence, and Brian McDonald, Rebecca Schwantes, Siyuan Wang for engaging in discussions at project meetings. We appreciate use of the emission inventories provided by Modeling and Meteorology Branch at CARB and NOAA Chemical Sciences Laboratory. We acknowledge the help from pilots, Bryce Kujat and George Loudakis, during the RECAP field campaign.

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
