# Peer review of "Direct observations of NOx emissions over the San Joaquin Valley using airborne flux measurements during RECAP-CA 2021 field campaign"

_Atmospheric Chemistry and Physics, 2023_

## Referee Comment (RC1)

Review of "Direct Observations of NOx emissions over the San Joaquin Valley using airborne flux measurements during RECAP-CA 2021 field campaign," Zhu et al., ACP (2023)

**Summary**

This manuscript presents observations of NOx emissions in the SJV acquired from an aircraft in summer 2021. They present the wavelet and footprint analysis, discuss weekday/weekend effects in total NOx emissions, and compare to inventories for urban, highway, and soil NOx emissions. They find that observed fluxes match well with inventories in urban areas but are higher than the models for highways and soils. The writing is good and the number and style of figures is appropriate. This paper fits well within the scope of ACP and is recommended for publication after considering the following minor revisions.

I further suggest that this paper be submitted as a potential EGU highlight paper.

**Specific Comments**

L10: is this bias within measurement uncertainty? If so, is it fair to call it a bias?

L16: Please clarify what is meant by "state-of-the-science model." Presumably these are the models that were compared with data in this paper? Or is something better needed?

L84: This instrument sounds fundamentally different to the instrument in the references (e.g., different laser with non-resonant detection). Some additional details are warranted, such as sample cell pressure and residence time, detection limit and/or precision.

L92: Do HRRR BL depths generally agree with aircraft vertical profiles?

L122: presumably dividing by std(NOx) was only done for the plot and not for the actual CWT calculation?

L127: the choice to smooth and re-sample fluxes is peculiar. It implies that individual data points are not independent. How do you propagate uncertainty in this case? Is the choice of 500 m effective horizontal resolution based on the cross-wind footprint width?

Figure 1 caption: "9th" should be "90th". Also, how is the flux detection limit defined?

L188: Could these high-tail fluxes also be real? For example, diesel trucks, offroad vehicles, or some intermittent high emitter? "incomplete sampling of eddies" seems like conjecture and only holds up near the edges where COI is important.

L221: This may be true if isolating the flux to the road land area; however, a model at 1 km resolution is also going to smear these out.

L227: This is the first mention of an emission inventory, but there is not description of said inventory in the methods or SI.

Sect. 5.2 details comparison to 3 different NOx emission schemes. All of these schemes are empirical (I think). It would be worth mentioning similarities and differences here. It is also maybe worth noting that the Guo 2020 study used DNDC, a fundamentally different biogeochemical model.

Fig. 7a: could offroad vehicles contribute to the large variability of RECAP "soil" NOx emissions?

L303: do MEGAN and BEIS not have a temperature dependence? If so, should be stated unambiguously.

L303ish: what about soil moisture dependence? Was there no rain during RECAP?

L314: If you are suggesting that MEGAN and BEIS parameterizations are not "state-of-the-science," you should do so explicitly. BDISNP also under-predicts.

Conclusions could benefit from an additional paragraph about limitations of the study (e.g., you didn't dig into soil moisture dependence) and what additional data is needed to reduce uncertainties in soil NOx modeling.

**Technical Comments**

L6: are these units mg of N? NO?

L100: recommend citing Torrence and Compo here.

L111: replace "largest" with "smallest" or "fastest"

Figure 3b: might be worth a log scale for the x-axis? Or both axes? Depends on what you want to highlight.

Fig. 4: should flux uncertainties appear here too?

Fig. 5: cannot distinguish red and blue when printing in greyscale. Recommend changing the hue of the red block to be lighter.

Fig. 7b: Recommend swapping axes and including the fit intercept and r^2.

L300: probably better to state the temperature range objectively without calling it "large variation."

Fig. S6: units for x-axis? Km?

---

## Referee Comment (RC2)

Review of "Direct Observations of NOx emissions over the San Joaquin Valley using airborne flux measurements during RECAP-CA 2021 field campaign," Zhu et al., ACP, 2023.

This paper presents measurements of NOx emissions made from an aircraft using the eddy covariance technique, over various land uses in the San Joaquin Valley, California in summer 2021. It presents comparisons of NOx emissions from highway, urban and soil with estimates from bottom up inventories. It is a generally excellent paper, well written and within the subject matter for ACP. I do have some concerns that should be addressed before publication:

**General comments:**

**Section 2:**

There is very little description of the instrument performance in this section. The authors have reference previous papers for the same instrument, but they are all relatively old. Some comment should be made of sensitivity and, in particular time, resolution of the instrument, with evidence needed that it is truly providing a 5Hz measurement. This is particularly important for the NO measurement, that is a result of the chemical transformation of NO to NO2 on addition of ozone. Is the chemistry fast enough to result in a 5Hz measurement?

**Section 3.4:**

The vertical divergence section needs expanding. The authors take a subset of all flux data measured over croplands to provide a homogeneous emission set of to use for vertical divergence assessment. There are a few questions about this methodology that need expanding upon. Do the croplands subset cover all different conditions experienced during the campaign and hence give a full picture of the flux divergence? How can the authors be sure that there are no other emission sources in the subset? For example, what about farm machinery that could produce a large perturbation to the data?

In figure 2, the linear regression is fit through medians of the z/zi data subsets. However, the spread on this data is quite large, as shown by the standard deviations. (also, would a median absolute deviation be more appropriate here?) The linear regression does not take into account this spread, so some more discussion is needed as to how this linear divergence feeds into the overall error calculation in the next section. With corrections up to 30% of the measured flux, it is important that this paid more attention.

**Section 3.5**

The authors note that they expect limited high-frequency loss due to the 5 Hz time resolution of the instrument. However, no quantification of this is presented – especially with the aforementioned concerns regarding the NO -> NO2 conversion rate, the authors should include estimates of the losses for each species.

Line 172 Here several papers that do not include an estimate of the uncertainty due to vertical flux divergence are listed. The wording is unclear a first reading as to whether they do not address the uncertainty, or do not address vertical flux divergence as a whole.

**Section 5:**

The first paragraph of this section would benefit from some discussion on how the measured emissions here compare to other similar studies. For instance, on line 243 it is stated that there are 'high' NOx emissions in the cities. Could the actual numbers be stated and maybe compared to other measurements from the literature?

**Section 6:**

The conclusions section needs further expanding to put the results in context. For instance, how important is the discrepancy between the measured and inventory NOx emissions in the area for regional air pollution models in the context of ozone formation. Clearly an order of magnitude underestimation of NOx emissions from soils is likely to be globally important so some comment should be made on this. Also, the authors should comment on the limitations of the study, especially with respect to whether their measured emissions are truly only coming from soil.

**Figures:**

The figures could be improved so that they all have similar formatting. Figures 1-4 look very different to 5-8 (font type, size, label size etc). There are also thick black borders around 5-8 which are not around 1-4. Figure 6 could also benefit from being shown as a scatter plot rather than box / whiskers.

**Specific:**

Section 3.3: It is not clear what the grid cell resolution is – please state it here.

Figure 1 (d): The map is very unclear. The labels are too small to read properly and the grey lines showing the percentiles of the footprint are too light. Also, I assume the caption is meant to say 90th percentile rather than 9th?

Line 214: What does k denote in the equation and why is it 1, 2 or 3. Presumably these are the land mass types but this should be stated.

Line 226: I'm not sure of the meaning of the phrase 'spatial resolution of the landscape'. Please expand as to what this means.

Line 287: What do the authors mean by a 'wide spread'? Please state explicitly what the spread of data is.

Line 238: What do the authors mean by 'significantly lower'? Please state the statistics used to come up with this statement.

Line 277: Capital B needed on biogenic.

---

## Author Comment (AC1)

**Response to Editor and Reviewers**
**"Direct observations of $NO_x$ emissions over the San Joaquin Valley using airborne flux measurements during RECAP-CA 2021 field campaign"**

Qindan Zhu[1,*], Bryan Place[2,**], Eva Y. Pfannerstill[3], Sha Tong[4,5], Huanxin Zhang[4], Jun Wang[4], Clara M. Nussbaumer[1,6], Paul Wooldridge[2], Benjamin C. Schulze[7], Caleb Arata[8], Anthony Bucholtz[9], John H. Seinfeld[7], Allen H. Goldstein[3,8], and Ronald C. Cohen[1,2]

[1]Department of Earth and Planetary Sciences, University of California, Berkeley, Berkeley, CA 94720, United States
[*]Now at Department of Earth, Atmospheric and Planetary Sciences, Massachusetts Institute of Technology, Cambridge, MA, United States of America
[2]Department of Chemistry, University of California, Berkeley, Berkeley, CA 94720, United States
[**]Now at Office of Research and Development, U.S. Environmental Protection Agency, Research Triangle Park, North Carolina 27711, United States
[3]Department of Environmental Science, Policy, and Management, University of California, Berkeley, Berkeley, CA 94720, United States
[4]Department of Chemical and Biochemical Engineering, Center for Global and Regional Environmental Research, and Iowa Technology Institute, University of Iowa, Iowa City, Iowa 52242, United States
[5]Collaborative Innovation Center on Forecast and Evaluation of Meteorological Disasters (CIC-FEMD), Key Laboratory for Aerosol-Cloud-Precipitation of China Meteorological Administration, Nanjing University of Information Science Technology, Nanjing 210044, People's Republic of China
[6]Department of Atmospheric Chemistry, Max Planck Institute for Chemistry, Mainz 55128, Germany
[7]Department of Environmental Science and Engineering, California Institute of Technology, Pasadena, CA 91125, United States
[8]Department of Civil and Environmental Engineering, University of California, Berkeley, Berkeley, CA 94720, United States
[9]Department of Meteorology, Naval Postgraduate School, Monterey, CA 93943, United States

**Correspondence:** Qindan Zhu (qindan_zhu@berkeley.edu) and Ronald C. Cohen (rccohen@berkeley.edu)

We thank all reviewers for their careful reading and comments. We also received some private comments that are attached at the end. The suggestions offered and questions raised are well taken, and we have done our best to incorporate them into the paper.

Below we respond to both common comments and individual comments. The reviewer's comments will be shown in red, our response in blue, and changes made to the paper are shown in black block quotes. Unless otherwise indicated, page and line numbers correspond to the original paper. Sections, figures, tables, or equations referenced as "R$n$" are numbered within this response; Figures, tables, and equations numbered normally refer to the numbers in the original discussion paper.

**R1 Review response to common comments**

**Before we address the individual comments, we would like to address some common comments from reviewers in three aspects.**

**R1.1 Unit correction**

First, thanks to the first reviewer, we realize an underlying error in our flux calculation. Our $NO_x$ flux, in units of mg m$^{-2}$ hr$^{-1}$, is calculated using the time series of $NO_x$ concentration in units of mg m$-3$, the sum of NO in units of mg m$^{-3}$ and $NO_2$ in units of mg m$^{-3}$. As a consequence, the calculated $NO_x$ flux, as well as the estimated $NO_x$ emissions, depends on the ratio of NO and $NO_2$. To correct this error, we only account for the mass of nitrogen in both NO and $NO_x$ concentration, and change the unit of $NO_x$ concentration to mg N m$^{-3}$, and the units of $NO_x$ flux and $NO_x$ emissions to mg N m$^{-2}$ hr$^{-1}$. Following the unit change, we redo the analysis and remake all the figures. It does not affect the majority of the conclusions but change all numbers we show in the context.

**R1.2 Interference of off-road vehicle emissions**

Second, we agree with the reviewers that there could be interference of soil $NO_x$ emission from off-road vehicle emissions. To address this interference, we make use of the trimethylbenzene flux observed during RECAP described in Pfannerstill et al. (2023), and interpolate them to the time series of $NO_x$ flux. We then categorize trimethyl benzene flux into two groups based on whether the footprint fully covers croplands or not, and the trimethyl benzene flux distributions of the two groups are shown in Figure R1. Overall, we see a much lower trimethyl benzene flux from croplands than from highway/urban land types, the median flux is 0.003 and 0.008 mg m$^{-2}$ hr$^{-1}$. The off-road vehicle emissions over soil land cover types are identified by a trimethyl benzene flux larger than 0.02 mg m$^{-2}$ hr$^{-1}$, and we remove the corresponding $NO_x$ flux observations in the calcuation of soil $NO_x$ emission. We add Figure R1 into the supplement, and include the discussion of filtering of $NO_x$ flux using trimethylbenzene flux in Sect. 3.4:

"It is worth noting that croplands includes not only soil $NO_x$ emissions but the off-road vehicle emissions. Erroneously attributing the $NO_x$ from off-road vehicle emissions to soil $NO_x$ emissions leads to a high bias. While trimethylbenzene was observed during RECAP-CA field campaign, Pfannerstill et al. (2023) presented the trimethylbenzene fluxed using the same algorithm described in Sect. 3.2. The trimethylbenzene fluxes are interpolated to match the $NO_x$ fluxes in time and are utilized as an indicator of off-road vehicle emissions over croplands (Tsai et al., 2014). The trimethylbeneze fluxes are categorized into two groups; the first group presents footprints covering croplands exclusively and the second group presents footprints with mixed land cover types. Shown in Figure S9, the trimethylbeneze flux is much lower over croplands, a median of 0.003 mg m$^{-2}$ h$^{-1}$ compared to a median of 0.009 mg m$^{-2}$ h$^{-1}$ over mixed land cover types including highway and urban areas. Among all observations over cropland, we identify those with the trimethylbeneze flux larger than 0.02 mg m$^{-2}$ h$^{-1}$, which consists of 7% of the total data points, are impacted by the off-road vehicle emissions, and then filter out them in the later analysis. We

[Figure]

**Figure R1.** a) the comparison of trimethylbenzene fluxes with footprints exclusively covering soil land cover type and those with footprints covering mixed land cover types. b) The distribution of trimethylbenzene fluxes with footprints exclusively covering soil land cover type. The dashed line denotes a trimethylbenzene flux of 0.02 mg m$^{-2}$ h$^{-1}$.

also vary the threshold of the trimethylbeneze flux between 0.005 mg m$^{-2}$ h$^{-1}$ and 0.04 mg m$^{-2}$ h$^{-1}$ and conclude that the choice of the threshold does not influence the results."

**R1.3 Uncertainty analysis**

Third, we acknowledge the weakness in our calculation of uncertainty. The detection limit and the uncertainty in the vertical divergence are not properly accounted for in the uncertainty analysis. Here, we calculate the flux detection limit following Langford et al. (2015) and provide a more thorough uncertainty analysis in Sect 3.5:

[revised manuscript text omitted]

„

**R2  Review response to Reviewer 1**

This manuscript presents observations of NOx emissions in the SJV acquired from an aircraft in summer 2021. They present the wavelet and footprint analysis, discuss weekday/weekend effects in total NOx emissions, and compare to inventories for urban, highway, and soil NOx emissions. They find that observed fluxes match well with inventories in urban areas but are higher than the models for highways and soils. The writing is good and the number and style of figures is appropriate. This paper fits well within the scope of ACP and is recommended for publication after considering the following minor revisions. I further suggest that this paper be submitted as a potential EGU highlight paper.

Thanks so much for the positive feedback and we are really grateful for the nomination of EGU highlight paper.

L10: is this bias within measurement uncertainty? If so, is it fair to call it a bias?

We agree that it is not fair to call it a bias. The new results show that the $NO_x$ emission derived from measured urban regions is 0.37 mg N m$^{-2}$ h$^{-1}$ on average, and it agrees quite well with both EMFAC and FIVE, which the mean urban $NO_x$ emission are 0.40 and 0.43 mg N m$^{-2}$ h$^{-1}$, respectively. Hence we changes L10:

100     "We show that two anthropogenic inventories for mobile sources, EMFAC (EMssion FACtor) and FIVE (Fuel-based Inventory for Vehicle Emissions)  **yield strong agreement with emissions derived from measured fluxes over urban regions.**"

L16: Please clarify what is meant by "state-of-the-science model." Presumably these are the models that were compared
105     with data in this paper? Or is something better needed?

We acknowledge that "state-of-the-science" is not the best wording here. We refer the "state-of-the-science" model to the biogeochemical process-based model for representing soil $NO_x$ emissions:

    "We conclude that soil $NO_x$ is a key feature of the $NO_x$ emissions in the SJV and that a  **biogeochemical process-based model** model of these emissions is needed to simulate emissions for modeling air quality in the region."

110     L84: This instrument sounds fundamentally different to the instrument in the references (e.g., different laser with non-resonant detection). Some additional details are warranted, such as sample cell pressure and residence time, detection limit and/or precision.

We agree with the reviewer that additional details are needed for the instrument. We expand the last paragraph of the Sect. 2:

[revised manuscript text omitted]

L92: Do HRRR BL depths generally agree with aircraft vertical profiles?

155 Good point. As we conducted the measurements of aircraft soundings at the beginning and end of the voyage in the SJV and before the stacked racetracks, we can estimate the PBL heights using the sharp gradient in the dew point, water concentration, benzene concentration and temperature, and interpolate the PBL heights to the full duration of the flight. We compare the PBL heights from HRRR against the extrapolated PBL heights, and the relative difference between HRRR and extrapolated PBL heights is shown in Figure R3. Overall, they agree well with each other, with $< 30\%$ difference for 80% of the measurements. 160 The largest difference is found for the flight leg in the Sierra Nevada foothills with complex topography.

[Figure]

**Figure R3.** The distribution of relative difference between PBL height from HRRR and interpolated PBL height from the measurements.

In our revision, We decide to switch the PBL heights from HRRR to interpolated PBL from the measurement. Besides, we account for the uncertainty of the PBL height of 30% into the calculation of $NO_x$ flux uncertainty in Sect 3.5 (see my response in Sect. R1.3).

L122: presumably dividing by std(NOx) was only done for the plot and not for the actual CWT calculation?

165     Sorry for the confusion, the detrending is done before the CWT calculation, the calculated flux is the integral of local wavelet co-spectra from the CWT times the variance of both time series. We modified the equations and the context between Line 115 to Line 125 for clarification:

    "For two simultaneous time series of $NO_x$ ($W_c(a,b)$) and vertical wind speed ($W_w(a,b)$), we **first detrend them by subtracting out the average followed by dividing the standard deviation of a scalar time series. Then we**  obtain the

170     wavelet cross-spectrum following Eqn. 9. The Morlet wavelet-specific reconstruction factor $C_\delta$ is 0.776. We then sum up over

the full frequency scales to yield a time series of flux (Eqn. 10).

$$\bar{W}_c = \frac{1}{N}\sum_{i=1}^{N} W_{ci} \quad \text{and} \quad \bar{W}_w = \frac{1}{N}\sum_{i=1}^{N} W_{wi} \tag{6}$$

$$\hat{\sigma}_c^2 = \frac{1}{N-1}\sum_{i=1}^{N}(W_{ci}-\bar{W}_{ci})^2 \quad \text{and} \quad \hat{\sigma}_w^2 = \frac{1}{N-1}\sum_{i=1}^{N}(W_{wi}-\bar{W}_{wi})^2 \tag{7}$$

$$W_c'(a,b) = \frac{(W_c(a,b)-\bar{W}_c)}{\hat{\sigma}_c} \quad \text{and} \quad W_w'(a,b) = \frac{(W_w(a,b)-\bar{W}_w)}{\hat{\sigma}_w} \tag{8}$$

175 $$E_{c,w}(j) = \frac{\delta t}{C_\delta}\frac{1}{N}\sum_{n=0}^{N-1}[W_c'(a,b)\cdot W_w'^*(a,b)] \tag{9}$$

$$F(t) = \hat{\sigma}_c\hat{\sigma}_w\overline{c'w'} = \hat{\sigma}_c\hat{\sigma}_w\frac{\delta t}{C_\delta}\frac{\delta j}{N}\sum_{n=0}^{N-1}\sum_{j=0}^{J}\frac{[W_c'(a,b)\cdot W_w'^*(a,b)]}{a(j)} \tag{10}$$

Figure 1 exhibits an example of CWT flux calculation. Figure 1 (a) shows the detrended NOx and vertical wind speed in a straight segment of $\sim$ 50 km. . ..."

180     L127: the choice to smooth and re-sample fluxes is peculiar. It implies that individual data points are not independent. How do you propagate uncertainty in this case? Is the choice of 500 m effective horizontal resolution based on the cross-wind footprint width?

    A moving average of 2km is applied to remove the flux perturbation due to local turbulence, which the spatial scale is generally less than 2km in the atmosphere. The 500m spatial average is consistent with the resolution of footprint width. We

185 add the clarification in Sect. 3.3:

    "**For each flux observation, we calculate the footprint map at the spatial resolution of 500m and then extract the 90% contour.**"

    Propagating the uncertainty can be tricky while two layers of averaging is applied to the datasets so that the data points are not independent. Therefore, when we propagate the uncertainty, we ignore the 2km moving average and assume the original

190 data sets are only spatially averaged to 500m in Sect. 3.5:

    "**Random errors arise from the noise in the instrument ($RE_{noise}$) as well as the noise in turbulence sampling ($RE_{turb}$), which are calculated using Eqn. 11 and Eqn. 12 (Wolfe et al., 2018; Lenschow et al., 1994)**.

$$RE_{noise} = \frac{\sqrt{\sigma_{NO_x,noise}^2\sigma_w^2}}{N} \tag{11}$$

$$\frac{RE_{turb}}{F} \leq 1.75(\frac{z}{z_i})^{0.25}(\frac{z_i}{L})^{0.5} \tag{12}$$

195      $z$, $L$ and $z_i$ are the same as Eqn. 1, $\sigma_w^2$ is the variance of vertical wind speed. **Note that $RE_{noise}$ assumes the noise in each time step uncorrelated, therefore, we ignore the moving average step in the uncertainty calculation and $N$ denotes the number of points used to yield each 500m spatially averaged flux.**"

Figure 1 caption: "9th" should be "90th". Also, how is the flux detection limit defined?

Thanks for identifying the typo. We change the caption of Figure 1:

200      "The gray lines show the  **90th** percentiles of the footprints and the black line denotes the contours of all footprints."

L188: Could these high-tail fluxes also be real? For example, diesel trucks, offroad vehicles, or some intermittent high emitter? "incomplete sampling of eddies" seems like conjecture and only holds up near the edges where COI is important.

We agree that the interference of off-road vehicle needs to be considered seriously. Therefore, we decide to use the trimethyl-benzene flux to identify the $NO_x$ fluxes impacted by the off-road vehicle emission and remove them in further analysis. Please

205      refer to our response in Sect. R1.2.

After removing the $NO_x$ fluxes impacted by the off-road vehicle emission, there is still high-tail fluxes. We think that it can be caused by incomplete sampling of eddies because our $NO_x$ concentration is at 5 Hz and the CWT cannot resolve the turbulence with the frequency less than 0.4 s (Sect 2.2).

L221: This may be true if isolating the flux to the road land area; however, a model at 1 km resolution is also going to smear

210      these out.

Correct. In this section, we would like to provide the best estimate of $NO_x$ component flux from highways, not yet the representation of this emission in the model framework.

L227: This is the first mention of an emission inventory, but there is not description of said inventory in the methods or SI.

We rephrase the sentence in L227:

215      "While these separate component fluxes emphasize the distinction between individual land types at the spatial resolution of the landscape, **we utilize the NOx fluxes to yield an estimate of NOx emission at 4km.**  "

Sect. 5.2 details comparison to 3 different NOx emission schemes. All of these schemes are empirical (I think). It would be worth mentioning similarities and differences here. It is also maybe worth noting that the Guo 2020 study used DNDC, a

220      fundamentally different biogeochemical model.

Yes, all 3 $NO_x$ schemes are empirical and the estimated soil $NO_x$ emission is parameterized. We compare these $NO_x$ schemes due to their wide application to the chemical transport model. We add the discussion at the beginning of Sect 5.2:

"Soil NO$_x$  **Soil NO$_x$ emissions are determined by biogeochemical processes including soil microbe-mediated nitrification and denitrification. Process-based biogeochemical models have been developed to mechanistically represent soil NO$_x$ emissions by simulating nitrogen interactions in ecological systems, such as DeNitrification-DeComposition (DNDC) (Li et al., 1992, 1994; Guo et al., 2020) and DayCENT (Del Grosso et al., 2000; Rasool et al., 2019). However, these process-level models are not yet widely applied to chemical transport models, and the default model configuration uses empirical soil NO$_x$ schemes.** The Model of Emissions of Gases and Aerosols from Nature v3 (MEGAN)..."

Fig. 7a: could offroad vehicles contribute to the large variability of RECAP "soil" NOx emissions?

Please refer to our resposne in Sect. R1.2.

L303: do MEGAN and BEIS not have a temperature dependence? If so, should be stated unambiguously.

We do not see the soil temperature dependence in either MEGAN or BEIS. Both MEGAN and BEIS estimate the soil NO$_x$ emission (Fi) based on emission factors and activity factors, and activity factors only account for ambient temperature, not soil temperature. The BDISNP parameterization is different from the MEGAN implementation and includes a better parameterization of meteorological conditions, including soil temperature, soil moisture, and fertilizer application.

L303ish: what about soil moisture dependence? Was there no rain during RECAP?

There is no rain during the duration of RECAP in June 2021. Soil moisture is for sure an important parameter affecting soil NO$_x$ emissions, the airborne measurements presented in this study are unable to tackle it due to limited variation in the soil moisture. Future study is needed to study the soil moisture dependence. SJV farm water is almost entirely irrigation controlled between April and November. We do not have detailed information on irrigation.

L314: If you are suggesting that MEGAN and BEIS parameterizations are not "state-of-the-science," you should do so explicitly. BDISNP also under-predicts.

Thanks for the suggestion. We acknowledge that "state-of-the-science" is not the best wording here. As we refer to the "state-of-the-science" model to the process-based biogeochemical model for representing soil NO$_x$ emissions and make changes in L314:

"However, the widely used, but not  **biogeochemical process-based**, models for soil NO$_x$ emissions underestimate"

Conclusions could benefit from an additional paragraph about limitations of the study (e.g., you didn't dig into soil moisture dependence) and what additional data is needed to reduce uncertainties in soil NOx modeling.

In terms of reducing uncertainty in soil NO$_x$ modeling, we end up using trimethylbeneze flux in this revision. Besides, We agree with the reviewer and add a discussion of the limitation of this study in Sect 5.3:

"**It is worth noting the limitation of estimated soil NO$_x$ emissions in our study. First of all, we are unable to investigate the dependence of soil NO$_x$ emissions on meteorological drivers other than soil temperature, such as soil moisture, as modeled soil moisture presents very small variation during the field campaign. Second, as our measurements only cover limited cropland areas in SJV over a short time period and it is around the time of fertilizer use, we cannot scale the estimated soil NO$_x$ emission to the whole year or to the total cropland areas in California. Last, in the absence of ozone and PM2.5 observations, we cannot investigate the impact of soil NO$_x$ emission on air quality. However, as the SJV is in the NO$_x$ limited regime (Pusede et al., 2014), we expect a model that captures the soil NO$_x$ more accurately will produce higher ozone. Future work is needed to further advance our understanding of soil NO$_x$ emission and its role in urban and rural air pollution.**"

L6: are these units mg of N? NO?

Thanks for pointing it out. We now change the unit to mg of N. Please refer to our response in Sect. R1.1.

L100: recommend citing Torrence and Compo here.

Thanks for the reminder. The citation is added:

"The continuous wavelet transformation (CWT) parameterization decomposes the time series ($x(t)$) into a range of frequencies and represents it as the convolution of the time series with a wavelet function (**Torrence and Compo, 1998**)"

L111: replace "largest" with "smallest" or "fastest"

It is changed to "smallest.""

"The  **smallest** frequency scale is the Nyquist frequency, which is twice the time resolution ($0.4s$)"

Figure 3b: might be worth a log scale for the x-axis? Or both axes? Depends on what you want to highlight.

The x-axis cannot be set to log scale due to the negative values. We would like to keep it as linear scales for both axes to highlight that 1) full range of the NO$_x$ flux and 2) and overall distribution. We think Figure 3b as well as the context provide a detailed statistical summary of NO$_x$ flux.

Fig. 4: should flux uncertainties appear here too?

Yes. The uncertainty of component flux accounts for the flux uncertainty by bootstrapping each flux observation within its uncertainty. In this revision, we switch to the Monte Carlo simulation to propagate the uncertainty. Overall, the uncertainty of component flux is small compared to the uncertainty of each flux observation.

Fig. 5: cannot distinguish red and blue when printing in greyscale. Recommend changing the hue of the red block to be lighter.

The figure is updated.

Fig. 7b: Recommend swapping axes and including the fit intercept and r2.

Figure 7 is updated by adding fit intercept and r. We decide to keep the axes the same as it is consistent with the statement "the model is 2.2 times lower than the observations" in the context.

L300: probably better to state the temperature range objectively without calling it "large variation."

We change the sentence in L300:

"**A range of soil temperature between 295K to 304K is observed.**  "

Fig. S6: units for x-axis? Km?

Correct, the figure is updated to include the unit (km).

**R3 Review response to Reviewer 2**

This paper presents measurements of NOx emissions made from an aircraft using the eddy covariance technique, over various land uses in the San Joaquin Valley, California in summer 2021. It presents comparisons of NOx emissions from highway, urban and soil with estimates from bottom up inventories. It is a generally excellent paper, well written and within the subject matter for ACP. I do have some concerns that should be addressed before publication:

Thanks so much for the positive feedback and the comments.

Section 2: There is very little description of the instrument performance in this section. The authors have reference previous papers for the same instrument, but they are all relatively old. Some comment should be made of sensitivity and, in particular time, resolution of the instrument, with evidence needed that it is truly providing a 5Hz measurement. This is particularly important for the NO measurement, that is a result of the chemical transformation of NO to NO2 on addition of ozone. Is the chemistry fast enough to result in a 5Hz measurement?

Thanks for pointing it out. We provide 4 seconds of reaction time before the detection of the second channel. The flow was continuous through all the tubing. If a step change in concentration enters at one end, then with ideal "plug flow" the same step change would exit the other end 4 seconds later, yielding $NO_x$ measurements every 0.2 s. Due to the additional reaction step, there is a time lag in the signals observed at the first and second channels. Therefore, we utilize the covariance of the

signals from both channels and correct the time shifts for the $NO_x$ measurements from the second channel. We expand the last paragraph of Sect. 2:

"The standard instruments aboard the aircraft are described in (Karl et al., 2013) and include total and dew point temperature, barometric and dynamic pressures, wind direction and wind speed, total airspeed, slip- and attack angles, GPS latitude, GPS longitude, GPS altitude, pitch, roll, and heading. These measurements are at 10 Hz temporal resolution. VOCs were measured at 10 Hz time resolution by Vocus proton transfer reaction time of flight mass spectrometer (Vocus PTR-ToF-MS) as described in Pfannerstill et al. (2023). Mixing ratios of $NO_x$ were measured at 5 Hz frequency using a custom-built three-channel thermal dissociation-laser induced fluorescence (TD-LIF) instrument. The multipass LIF cells, fluorescence collection, long-pass wavelength filtering (for $\lambda > 700$ nm), and photon counting details have been previously described (Thornton et al., 2000; Day et al., 2002; Wooldridge et al., 2010). Details specific to this implementation are described below.

Air was sampled from the aircraft community inlet through PFA Teflon tubing at a rate of ~6 L/min and split equally between the three instrument channels. Each measured $NO_2$ by laser-induced fluorescence utilizing a compact green laser (Spectra-Physics ExplorerOneXP 532 nm). The laser was pulsed at 80 kHz and the 1.7 Watt average power was split between the three cells. Earlier versions of the instrument used a dye laser tuned on and off a narrow rovibronic $NO_2$ resonance at 585.1 nm. Experience over a wide variety of conditions had demonstrated the off-line signal did not depend on the sample, other than from aerosol particles and that could be eliminated by adding a Teflon membrane filter. Moving to nonresonant excitation at 532 nm provided full-time coverage at 5 Hz along with lower complexity and more robust performance of the laser system. Maintaining the LIF cells at low pressure (~0.4 kPa) was no longer required to avoid line-broadening but was still desirable to extend the $NO_2$ fluorescence lifetime for time-gated photon counting to reject prompt laser scatter. Instrument zeros were run using ambient air scrubbed of $NO_x$ every 20 minutes in flight to correct for any background drift during the flights. In addition, calibrations were performed in-flight every 60 minutes using a $NO_2$ in $N_2$ calibration cylinder (Praxair, 5.5 ppm, Certified Standard grade) diluted with scrubbed air.

$NO_2$ was measured directly in the first channel, with the sample passing only through a particle filter and a flow-limiting orifice before the cell. $NO_x$ was measured in the second by adding $O_3$ (generated with 184.5 nm light and a flow of scrubbed and dried air) to convert NO to $NO_2$ before detection. A 122 cm length of 0.4 cm i.d. tubing served as the $O_3$+NO reactor, providing 4 seconds of reaction time before the orifice. The third channel was used to measure the sum of higher nitrogen oxides (e.g. organic nitrates and nitric acid) by thermal dissociation to $NO_2$ with an inline oven (~500 C) before LIF detection."

Section 3.4: The vertical divergence section needs expanding. The authors take a subset of all flux data measured over croplands to provide a homogeneous emission set of to use for vertical divergence assessment. There are a few questions about this methodology that need expanding upon. Do the croplands subset cover all different conditions experienced during the campaign and hence give a full picture of the flux divergence? How can the authors be sure that there are no other emission sources in the subset? For example, what about farm machinery that could produce a large perturbation to the data?

We use the subset of fluxes over croplands to calculate the vertical divergence while minimizing the interference of emission inhomogeneity. In terms of other emission sources over cropland, we decide to use the trimethylbenzene flux to identify the

340   NO$_x$ fluxes impacted by the off-road vehicle emission and remove them in further analysis. Please refer to our response in Sect. R1.2.

In figure 2, the linear regression is fit through medians of the z/zi data subsets. However, the spread on this data is quite large, as shown by the standard deviations. (also, would a median absolute deviation be more appropriate here?) The linear regression does not take into account this spread, so some more discussion is needed as to how this linear divergence feeds into

345   the overall error calculation in the next section. With corrections up to 30% of the measured flux, it is important that this paid more attention.

The median in each interval removes the scatter that is due to the scatter in individual surface fluxes and should result in the median vertical divergence, which is expected to be approximately constant for each respective region. There is no large difference between standard deviation and median absolute deviation. We also calculate the vertical divergence using the mean

350   flux in each interval and the resulting vertical correction factor is quite similar (Figure R4). We take it into account and calculate an uncertainty of 40% by bootstrapping the data points in each interval. Besides, we acknowledge that the uncertainty can be even larger given the choice of data subset and the interference of emission inhomogeneity. Therefore, we tentatively increase the uncertainty from 40% to 100% from the vertical divergence and propagate it into the total uncertainty. The changes are added in Sect. 3.5:

355   "Estimating the uncertainty caused by the correction of vertical divergence is tricky. While we conclude that the influence of vertical divergence is non-negligible, it is ignored in some previous airborne flux studies (e.g. Vaughan et al., 2016; Hannun et al., 2020; Vaughan et al., 2021; Drysdale et al., 2022). **While the flux is scattered in each vertical intervals in our divergence calculation, we first bootstrap the flux observations and calculate the uncertainty of correction factor ($\sigma_C$) to 40%. As we see a significant difference in vertical correction factor on racetrack measurements versus a selected**

360   **subset of flux observations, we tentatively set the uncertainty of $C$ to 100%, in order to account for the case of no vertical divergence. Besides, we account for a 30% uncertainty in the PBL heights.**"

Section 3.5 The authors note that they expect limited high-frequency loss due to the 5 Hz time resolution of the instrument. However, no quantification of this is presented – especially with the aforementioned concerns regarding the NO -> NO2 conversion rate, the authors should include estimates of the losses for each species.

365   The high-frequency loss means that the CWT can only diagnose the turbulence with the frequency higher than the Nyquist frequency, which is twice the time resolution. The limited high-frequency loss can be demonstrated by co-spectra of NO$_x$ in Figure R5 (also Figure S5 in the supplement).

The uncertainty from the chemical conversion is considered the noise in the instrument and is considered in the calculation of flux detection limit in Sect. 3.5:

370   "The flux detection limit does not only depend on the signal-to-noise ratio of the NO$_x$ measurement, but also varies with wind speed and atmospheric stability. Following Langford et al. (2015), we calculate the detection limit of flux (LoD) before the

[Figure]

**Figure R4.** Vertical profiles of measured fluxes above croplands during RECAP-CA field campaign binned by the ratio of measurement height and PBL height ($z/z_i$). The red and blue points represent the median and mean flux within each bin, respectively. The error bars represent the standard deviation. The red and blue dashed line show the linear fits for median and mean fluxes versus relative height.

moving and spatial average are applied. For each segment, the observed $NO_x$ is replaced with a white noise time series and is then feed into the CWT to yield the corresponding time series of "noise" flux. The random error affecting the flux ($\sigma_{NO_x,noise}$) is defined as the standard deviation of this noise-derived flux, and LoD is defined as $2 \times \sigma_{NO_x,noise}$ ($95^{th}$ confidence level).

375 Among 142 segments, Figure R2 (a) shows the distribution of flux LoD among 142 segments. The LoDs range from 0.02 mg N m$^{-2}$ h$^{-1}$ to 0.30 mg N m$^{-2}$ h$^{-1}$, and the average LoD is 0.10 mg N m$^{-2}$ h$^{-1}$. To obtain a better constraint on the flux quality, we compare the LoD against the time series of flux in each segment and filter out 18 segments in which the whole time series is below the LoD. "

[Figure]

**Figure R5.** Normalized co-spectra of $NO_x$ and heat flux. The black dashed represents the Nyquist frequency for $NO_x$ flux.

Line 172 Here several papers that do not include an estimate of the uncertainty due to vertical flux divergence are listed. The wording is unclear a first reading as to whether they do not address the uncertainty, or do not address vertical flux divergence as a whole.

We agree with the reviewer and decide to take the uncertainty from the vertical flux divergence into account. Please refer to our response in Sect. R1.3.

Section 5: The first paragraph of this section would benefit from some discussion on how the measured emissions here compare to other similar studies. For instance, on line 243 it is stated that there are 'high' NOx emissions in the cities. Could the actual numbers be stated and maybe compared to other measurements from the literature?

Good point. we include a new Sect 5.3 to compare the estimated soil $NO_X$ emissions against those in the field experiments: "**Soil $NO_x$ emissions in California have been studied in field experiments. Matson et al. (1997) measured soil $NO_x$ emissions from nine dominant crop types in SJV and reported mean fluxes of 0.01-0.09 mg N m$^{-2}$ h$^{-1}$. They also reported a large variation of measured $NO_x$ flux among crops and among different fields of the same crop; the highest measured $NO_x$ flux is 0.17 mg N m$^{-2}$ h$^{-1}$ due to the fertilizer application and soil moisture characteristics. Horwath and Burger (2013) observed an average flux of 0.05-0.28 mg N m$^{-2}$ h$^{-1}$ at mid-days during summertime from five crops in California, and the highest $NO_x$ flux is >4 mg N m$^{-2}$ h$^{-1}$ in systems receiving large N inputs resulting in high concentrations of ammonium. Oikawa et al. (2015) observed soil $NO_x$ emissions in a high-temperature fertilized agricultural region of the Imperial Valley, CA, ranging between -0.02 and 3.2 mg N m$^{-2}$ h$^{-1}$. They also conducted**

control experiments to investigate the soil $NO_x$ emission responses to fertilization and irrigation. The highest soil $NO_x$ flux was reported ~10 days after the fertilizer at the soil volumetric water content of 30% and the soil temperature of ~313K.

The mean soil $NO_x$ flux, 0.32 mg N m$^{-2}$ h$^{-1}$, derived in our flux measurements is higher than the mean fluxes reported in Matson et al. (1997) and Horwath and Burger (2013), however, the range of estimated soil $NO_x$ flux is within those in Horwath and Burger (2013) and Oikawa et al. (2015). Fertilizer is likely the primary contributor to the higher mean soil $NO_x$ flux in our study. The RECAP-CA field campaign was conducted in June, right after the month of peak fertilizer use in SJV (Guo et al., 2020). Shown in Oikawa et al. (2015), soil $NO_x$ flux can increase up to 5-fold within 20 days of fertilizer. The higher mean soil $NO_x$ flux is also contributed by higher soil temperature. In our study, the mean soil temperature is 299K with a range between 295K and 304K, whereas the observations in Horwath and Burger (2013) and Oikawa et al. (2015) spread over a wider range of soil temperature, 288K-315K. Consistent with our study, the temperature dependence of soil $NO_X$ emission is observed in these field experiments. Horwath and Burger (2013) reported a 2.5-3.5 fold increase in $NO_x$ fluxes with 10-degree increase in soil temperature. Oikawa et al. (2015) showed that the temperature dependence of soil $NO_x$ emission is non-linear; a steeper increase in soil $NO_x$ emission was observed with the soil temperature exceeding 295K. ”

In L243, the "high" emission over urban areas and highway 99 is the description of Figure 5(b) (darker red on these areas).

Section 6: The conclusions section needs further expanding to put the results in context. For instance, how important is the discrepancy between the measured and inventory NOx emissions in the area for regional air pollution models in the context of ozone formation. Clearly an order of magnitude underestimation of NOx emissions from soils is likely to be globally important so some comment should be made on this. Also, the authors should comment on the limitations of the study, especially with respect to whether their measured emissions are truly only coming from soil.

In the absence of ozone measurements during RECAP field campaign, we are unable to study the impact of soil $NO_x$ emissions on ozone formation. We considered it as one of the limitations of this study. In Sect 5.3, we add the discussion of the limitation of this study:

"It is worth noting the limitation of estimated soil $NO_x$ emissions in our study. First of all, we are unable to investigate the dependence of soil $NO_x$ emissions on meteorological drivers other than soil temperature, such as soil moisture, as modeled soil moisture presents very small variation during the field campaign. Second, as our measurements only cover limited cropland areas in SJV over a short time period and it is around the time of fertilizer use, we cannot scale the estimated soil $NO_x$ emission to the whole year or to the total cropland areas in California. Last, in the absence of ozone and PM2.5 observations, we cannot investigate the impact of soil $NO_x$ emission on air quality. However, as the SJV is in the $NO_x$ limited regime (Pusede et al., 2014), we expect a model that captures the soil $NO_x$ more accurately will produce higher ozone. Future work is needed to further advance our understanding of soil $NO_x$ emission and its role in urban and rural air pollution."

Figures: The figures could be improved so that they all have similar formatting. Figures 1-4 look very different to 5-8 (font type, size, label size etc). There are also thick black borders around 5-8 which are not around 1-4. Figure 6 could also benefit from being shown as a scatter plot rather than box / whiskers.

Thanks for pointing it out. The figures look different because we use Matlab to produce Figures 1-4 and use Python to produce Figures 5-8. We have updated the figures to make them look more synchronized. For Figure 6, we add the scatter plot into the supplement.

Section 3.3: It is not clear what the grid cell resolution is – please state it here. Figure 1 (d): The map is very unclear. The labels are too small to read properly and the grey lines showing the percentiles of the footprint are too light. Also, I assume the caption is meant to say 90th percentile rather than 9th?

Figure 1 (d) is the overlay of one flux segment and the footprint contours on the "street" basemap using the geoscatter function in Matlab. The resolution of the footprint is 500m. We improve the figure quality and change the color of footprint contours from gray to black. We are sorry for the typo and have fixed it.

""The  **black** lines show the  **90th** percentiles of the footprints and the thick black line denotes the contours of all footprints." "

Line 214: What does k denote in the equation and why is it 1, 2 or 3. Presumably these are the land mass types but this should be stated.

It is correct, the k denotes each land cover type. We change the wording here:

"where $k_1$ **to** $k_3$ **denote highway, urban, and soil land types,** $w_k$ is the fractional area within the 90% footprint contour and $F_k$ are the corresponding component fluxes ."

Line 226: I'm not sure of the meaning of the phrase 'spatial resolution of the landscape'. Please expand as to what this means.

It means the spatial resolution of land cover type, which is 500m:

"While these separate component fluxes emphasize the distinction between individual land types at the spatial resolution of  **the land cover (500m)**, we..."

Line 287: What do the authors mean by a 'wide spread'? Please state explicitly what the spread of data is.

We change the words and replace it with a more quantitatively description.

"The analysis of the observations exhibits a median cultivated soil $NO_x$ emission of 0.26 mg N m$^{-2}$ h$^{-1}$  **; the interquartile range of the inferred emission is 0.14 mg N m$^{-2}$ h$^{-1}$ and 0.45 mg N m$^{-2}$ h$^{-1}$.** "

Line 238: What do the authors mean by 'significantly lower'? Please state the statistics used to come up with this statement.

We decide to not report the estimated $NO_x$ emission during weekends since too limited observations are maintained after the data quality control is applied to filter out the segments below the LoD.

460 Line 277: Capital B needed on biogenic.

It is fixed.

"The  **Biogenic** Emission Inventory System (BEIS) ......"

**R4    Review response to Reviewer 3**

Having considerable experience with the topic at hand (ariborne NOx flux estimates from the San Joaquin Valley), I am
465 concerned by the authors' lack of reference to the considerable volume of prior work that has occurred before them on this very topic. However, mine is not only a complaint about the professional cordiality of such an oversight, it has direct technical bearing on the accuracy of the work.

We really appreciate the reviewer's comments and we hope to address the reviewer's concern by point-to-point response.

The main concern with the work involves the use of a model to determine the boundary layer depths in the Southern San
470 Joaquin Valley, a numerically challenging parameter given the complex topograpy and unconstrained irrigation patterns on the valley floor (Bianco et al, 2011; Faloona et al., 2020; Alexander et al., 2022; Caputi et al., 2022). All of these works (and other references below) show that average summertime boundary layer heights in the region are about 500-600 m agl, which means that the lowest flight leg ( 300m) would be at or above z/zi   0.5. Figure 2 of the submitted manuscript clearly shows that the authors are using boundary layer depths (from the HRRR model) that are 1,200 m or greater, which is twice as deep
475 as the typicall boundary layer in the Southern San Joaquin Valley. This dramatically alters their analysis, and should not be overlooked. Alexander et al. (2022), in particular, shows how WRF model boundary layer heights are often several hundred meters deeper than observed in this region.

We appreciate that the reviewer points out the weakness of PBLH from HRRR reanalysis. While the PBLH is not continuously measured during the flight, we conducted the measurements of aircraft soundings at the beginning and end of each flight
480 in SJV and before the stacked racetracks, we can estimate the PBL heights using the sharp gradient in the dew point, water concentration, benzene concentration and temperature, and then interpolate the PBL heights to the full duration of the flight. Pfannerstill et al. (2023) has showed the range of PBL height, which is derived from the aircraft measurement, in each flight day during RECAP-CA field campaign. The lowest PBL height is 826m ($\pm$ 126m) on June 22th and the highest PBL height is 1303 ($\pm$ 114m) on June 3rd. Our calculated PBL height is well above 500-600 agl as the reviewer suggested.

485     We also compare the PBL heights from HRRR against the interpolated PBL heights, and the relative difference between HRRR and extrapolated PBL heights is shown in Figure R3. Overall, it agrees well with each other, $< 30\%$ difference for 80% of the measurements. The largest difference is found for the flight leg in the Sierra Nevada foothills with complex topography.

    In response to reviewer's concern on HRRR PBLH, we decide to switch the PBLH used in our analysis from HRRR to the interpolated PBL height from aircraft measurements. Besides, we consider an uncertainty of 30% from the PBLH and
490 propagate it into the total flux uncertainty (please refer to our response in Sect. R1.3).

    At issue is the NOx flux divergence across the depth of the boundary layer because the measurements at the lowest aircraft flight altitude have to be extrapolated to the surface. In Line 142 the authors erroneously state that, "The vertical divergence is a result of multiple processes, including net in-situ production or loss, storage, and horizontal advection." I believe this statement to be misleading. These are all terms in the mean scalar (NOx) budget equation, but one term does not necessarily result from
495 the action of the other terms directly. These terms are *not* in the governing equation for a scalar flux divergence, which can be derived. For example, net photochemical production will not "cause" a flux divergence if it is evenly distributed within the ABL, or if the time scale of production is much longer than the turbulent mixing (Damköhler number $\ll 1$).

    From the perspective of the boundary layer dynamics, the flux divergence is to first order the consequence of a surface flux (what is trying to be measured) and the entrainment flux. The latter can be estimated from the knowledge of the jump in [NOx]
500 across the top of the boundary layer and the many reports of midday entrainment now established in the literature (Karl et al., 2013; Trousdell et al., 2016; Trousdell et al., 2019; Caputi et al., 2022). For this reason, I would recommend showing profiles of mean flight data for water vapor, potential temperature, NOx, and turbulent kinetic energy (or $<w2>$) to determine where the boundary layer top is, and then to estimate the entrainment flux of NOx, which will be positive due to dilution. Because NOx is a scalar similar to water vapor because they both have a source at the surface and a dilution flux at the top, it could very well
505 mean that the flux divergence is minimal and the extrapolation will not lead to such a large change in flux from that measured by the aircraft.

    We respectfully disagree with the reviewer on this point. Entrainment at the boundary layer top is not the only cause of the vertical divergence, and it is technically included in the net in-situ loss term. The same statement can be found in many studies on flux analysis, such as Karl et al. (2013); Wolfe et al. (2015, 2018). For instance, Misztal et al. (2014) showed that while the
510 vertical divergence of isoprene flux is primarily controlled by chemical loss, the contribution of the storage term is the order of 2–5 %. Pigeon et al. (2007) showed that the main origin of vertical divergence is horizontal advection. While each component contributing to the vertical divergence can be quantified, we use the flux measurements at multiple altitudes in the boundary layer as a means of directly quantifying the flux divergence slope.

    Extrapolating Figure 2 up to $z_i$ shows a downward flux of about -0.5 mg/m2/hr (indicating a source of NOx above the top
515 of the ABL), which is further evidence that the $z_i$ being used is erroneous. Moreover, the exclusion of 3 out of the 7 flux data

[Figure]

**Figure R6.** Vertical profiles of measured fluxes above croplands during RECAP-CA field campaign binned by the ratio of measurement height and PBL height ($z/z_i$). The points represent the median flux within each bin, and the error bars represent the standard deviation. The red dashed line shows a linear fit for median fluxes versus relative height.

points (top one and bottom two in Fig. 2) in fitting the line to get the flux divergence is not rationalized and appears arbitrary and incorrect.

It is a very good point. However, we do not think it is evidence of erroneous zi. Instead, as we apply a stricter data quality control based on the flux detection limit (please refer to our response in Sect. R1.3) and trimethylbeneze flux (please refer to our response in Sect. R1.2), Figure 2 (also Figure R6 in this response) is greatly improved. we are not excluding any data points in the linear fit. With the updated Figure 2, extrapolating the line up to zi shows a flux of -0.09 mg N m$^{-2}$ h$^{-1}$, which does not indicate a source of NO$_x$ but the uncertainty in our calculation. It is further confirmed by bootstrapping the data points in each interval and calculating the intercept from the linear fit. The intercept is -0.05 ($\pm$ 0.08 mg N m$^{-2}$ h$^{-1}$).

At the very least I believe consideration of this one very important point about the boundary layer depths should be addressed before proceeding to publication. Nevertheless, I will put my other comments in a separate section of this review.

We hope that our response and changes related to the PBL heights have addressed your concern.

We believe the authors do a disservice to the ongoing scientific debate by simply stating the results of Almaraz et al. (2018) and then stating that these are contradicted by a different study by Guo et al. (2020). Further evidence supporting the hypothesis

of much larger NOx emissions in the SJV than modeled by Guo et al. (2020) have been adduced in the works of Sha et al. (2021), Wang et al. (2021), and Luo et al. (2022).

We'd like to make the point that the contribution of soil $NO_x$ emissions from SJV is highly uncertain. Therefore, we only include the studies that report the contribution of soil $NO_x$ emissions to total $NO_x$ emissions in California. We still agree with the reviewer so we add Sha et al. (2021) in L40-43:

"**The contribution of soil $NO_x$ emissions remains highly uncertain. While Guo et al. (2020) attribute approximately 1.1% of anthropogenic $NO_x$ emissions in California to soil $NO_x$,** Almaraz et al. (2018) argued that due to growing N fertilizer use, the SJV has soil $NO_x$ emissions of 24 kg of N ha$^{-1}$ year$^{-1}$, contributing 20-51% of the $NO_x$ budget of the entire state of California. **Similarly, Sha et al. (2021) estimated that 40.1% of the total NOx emissions over California in July 2018 are from soils.** "

We also feel like there was a missed opportunity in comparing similar soil NOx flux estimates, such as those presented in the exact same area by Trousdell et al. (2019) (their Fig. 3) but in units of kg N ha-1 yr-1. Other urban emissions to compare to include those found in Karl et al. (2023) and Vaughan et al., (2021), but in units of nmol m-2 s-1.

Good point. We are concerned about scaling our estimated soil $NO_X$ fluxes to the whole year because our observations were conducted in June, right after the peak month of fertilizer use. Scaling it to the whole year using the soil $NO_X$ emission in June will lead to a very high bias. Instead, we compare our estimated soil $NO_X$ fluxes against studies reporting soil $NO_x$ flux in units of mg N m$^{-2}$ h$^{-1}$. The comparison result is added as Sect 5.3.

"Soil $NO_x$ emissions in California have been studied in field experiments. Matson et al. (1997) measured soil $NO_x$ emissions from nine dominant crop types in SJV and reported mean fluxes of 0.01-0.09 mg N m$^{-2}$ h$^{-1}$. They also reported a large variation of measured $NO_x$ flux among crops and among different fields of the same crop; the highest measured $NO_x$ flux is 0.17 mg N m$^{-2}$ h$^{-1}$ due to the fertilizer application and soil moisture characteristics. Horwath and Burger (2013) observed an average flux of 0.05-0.28 mg N m$^{-2}$ h$^{-1}$ at mid-days during summertime from five crops in California, and the highest $NO_x$ flux is >4 mg N m$^{-2}$ h$^{-1}$ in systems receiving large N inputs resulting in high concentrations of ammonium. Oikawa et al. (2015) observed soil $NO_x$ emissions in a high-temperature fertilized agricultural region of the Imperial Valley, CA, ranging between -0.02 and 3.2 mg N m$^{-2}$ h$^{-1}$. They also conducted control experiments to investigate the soil $NO_x$ emission responses to fertilization and irrigation. The highest soil $NO_x$ flux was reported ~10 days after the fertilizer at the soil volumetric water content of 30% and the soil temperature of ~313K.

The mean soil $NO_x$ flux, 0.32 mg N m$^{-2}$ h$^{-1}$, derived in our flux measurements is higher than the mean fluxes reported in Matson et al. (1997) and Horwath and Burger (2013), however, the range of estimated soil $NO_x$ flux is within those in Horwath and Burger (2013) and Oikawa et al. (2015). Fertilizer is likely the primary contributor to the higher mean soil $NO_x$ flux in our study. The RECAP-CA field campaign was conducted in June, right after the month of peak fertilizer use in SJV (Guo et al., 2020). Shown in Oikawa et al. (2015), soil $NO_x$ flux can increase up to 5-fold within 20 days of fertilizer. The higher mean soil

NO$_x$ flux is also contributed by higher soil temperature. In our study, the mean soil temperature is 299K with a range between 295K and 304K, whereas the observations in Horwath and Burger (2013) and Oikawa et al. (2015) spread over a wider range of soil temperature, 288K-315K. Consistent with our study, the temperature dependence of soil NO$_X$ emission is observed in these field experiments. Horwath and Burger (2013) reported a 2.5-3.5 fold increase in NO$_x$ fluxes with 10-degree increase in soil temperature. Oikawa et al. (2015) showed that the temperature dependence of soil NO$_x$ emission is non-linear; a steeper increase in soil NO$_x$ emission was observed with the soil temperature exceeding 295K. "

In Sect 5.3, we also describe the limitation of this study and show why we cannot scale the soil NO$_x$ emission from this study to the whole year or to the full cropland areas in California:

"It is worth noting the limitation of estimated soil NO$_x$ emissions in our study. First of all, we are unable to investigate the dependence of soil NO$_x$ emissions on meteorological drivers other than soil temperature, such as soil moisture, as modeled soil moisture presents very small variation during the field campaign. Second, as our measurements only cover limited cropland areas in SJV over a short time period and it is around the time of fertilizer use, we cannot scale the estimated soil NO$_x$ emission to the whole year or to the total cropland areas in California. Last, in the absence of ozone and PM2.5 observations, we cannot investigate the impact of soil NO$_x$ emission on air quality. However, as the SJV is in the NO$_x$ limited regime (Pusede et al., 2014), we expect a model that captures the soil NO$_x$ more accurately will produce higher ozone. Future work is needed to further advance our understanding of soil NO$_x$ emission and its role in urban and rural air pollution."

In terms of comparing urban emissions to other studies, we decide not to include it in the context. As we are intended to use NO$_x$ flux to evaluate the anthropogenic NO$_x$ emission inventory, the city-wise comparison of anthropogenic NO$_x$ emission is not in the scope of this work.

Line 127: What are you considering large-scale here? The cospectrum of Fig. 3S seems to indicate peaks near 2.2 km, and a secondary peak near 8.5 km, so why filter at 2 km?

We take 2km moving average because the largest length scale of turbulence within the boundary layer should be the PBL height, which is less than 2km.

Line 136: The Högström (1988) similarity functions apply only to the surface layer winds, not throughout the depth of the entire boundary layer.

Thanks for pointing it out. Instead of calculating frictional velocity using the log-wind profile, we decide to use the frictional velocity from the HRRR product.

"  **We obtain the surface roughness length and friction velocity from the HRRR product.** "

Line 185/189: The undersampling of turbulent eddies would lead to a systematic error, but it should not lead to a sign change (downward flux). Unless you are suggesting that some eddies are carrying downward fluxes, different than other sizes? Similarly, there is no reason to believe that incomplete sampling would lead to random large spikes in upward fluxes. Again, this is a function of instrument noise, or actually surface regions of high fluxes, not systematic errors due to frequency sampling limitations.

We attribute the negative flux to the large uncertainty in our flux calculation. The systematic error due to undersampling is one source of uncertainty, but we agree with the reviewer that it is not the major cause. We have conducted a more thorough uncertainty analysis in this revision, please refer to our response in Sect. R1.3. We also change the L185:

"We attribute the remaining 10% of negative fluxes to the uncertainties in the flux calculation "

Line 199: "due to higher altitude above the ground relative to the boundary layer height" is a statement that is difficult to understand. Are the winds stronger "at the foothills"? This is data that you should have measured. Are the "foothills" flight tracks at higher altitude than the others?

Yes the "foothills" flight tracks have higher altitudes than others and experience stronger winds. The flight was lifted as it got closer to the mountain. As a result, the ratio of flight altitude over the PBL heights is larger at the foothills. We change the L199 for clarification:

"The largest footprint extent corresponds to observations at the foothills, due to higher altitude  relative to the boundary layer height and stronger horizontal wind advection. "

**R5    Review response to Private Reviewer**

1. The calculated NOx emissions during the weekends are very low, as seen in Figure 7. The explanation is "we attribute to reduced diesel truck activity at the weekend". However, this also suggests that diesel truck emissions dominantly contribute to overall NOx emissions. The data on weekends also indicates that the typical soil NOx emission is about 0.01 mg/m2/h on average, which does not align with the calculated cropland NOx flux of 0.79 mg/m2/h from the multivariate analysis.

Thanks for spotting the difference. The low estimated $NO_x$ emission is predominantly due to the poor data quality on that day. As we apply a stricter data quality control and remove the segments with the full time series of flux below the detection limit, the majority of segments observed on June 12 are filtered out. Therefore, We decide to remove our estimated $NO_x$ emission on June 13 from Figure 5.

2. The authors did not use racetrack measurements to analyze vertical divergence because they found no clear pattern of fluxes increasing or decreasing with height due to variations in emissions. Instead, they employed a different method. However, as seen in Figure 2, this alternative approach also lacks a consistent pattern and some data points (lowest two points and the

highest point) deviate from the trend line. It is not clear why the mean value was not used (median value was used) or why all data points were not included in the linear regression analysis. The large error bars also make it difficult to determine a clear trend.

625    The reason for not using the racetrack or not using the full dataset is due to the emission inhomogeneity. The optimal approach to calculate the vertical divergence is to conduct stacked flux measurements at different altitude bins while the emissions within the footprints are the same. Otherwise we cannot attribute the vertical difference in the observed $NO_x$ flux solely to the vertical divergence. Unfortunately, our racetrack measurements are at the regions with mixed emission sources. Therefore, we decide to only select the observations whose footprints exclusively covering croplands to have the emission with 630    the smallest variation (compared to emission from highway).

With a stricter data quality control, we are now able to get a clearer trend of vertical divergence. We also calculate the vertical divergence using the mean flux in each interval and the resulting vertical correction factor is quite similar (Figure R4). We take it into account and calculate an uncertainty of 40% by bootstrapping the data points in each interval. Besides, we acknowledge that the uncertainty can be even larger given the choice of data subset and the interference of emission inhomogeneity. There- 635    fore, we tentatively increase the uncertainty from 40% to 100% from the vertical divergence and propagate it into the total uncertainty. The changes are added in Sect. 3.5:

"Estimating the uncertainty caused by the correction of vertical divergence is tricky. While we conclude that the influence of vertical divergence is non-negligible, it is ignored in some previous airborne flux studies (e.g. Vaughan et al., 2016; Hannun et al., 2020; Vaughan et al., 2021; Drysdale et al., 2022). **While the flux is scattered in each vertical intervals in our** 640    **divergence calculation, we first bootstrap the flux observations and calculate the uncertainty of correction factor ($\sigma_C$)** **to 40%. As we see a significant difference in vertical correction factor on racetrack measurements versus a selected** **subset of flux observations, we tentatively set the uncertainty of $C$ to 100%, in order to account for the case of no** **vertical divergence. Besides, we account for a 30% uncertainty in the PBL heights.**"

3. The authors did not provide an explanation as to why they reduced the resolution of the CropScape Database from 30-m 645    to 500-m. This change makes it more difficult to accurately estimate emissions from highways.

Sorry for the confusion. The selection of 500m spatial average is to match the spatial resolution of the footprint. Even though original resolution of CropScape is 30m, our calculated footprint cannot make distinction within 30m. We add the clarification in Sect. 3.3:

"**For each flux observation, we calculate the footprint map at the spatial resolution of 500m and then extract the 90%** 650    **contour.**"

4. In Figure 8, the relationship between temperature and the RECAP measurements is not very clear, as the variation is large and seems to increase at higher temperatures. The authors should perform statistical tests to determine if the differences between the three groups are meaningful.

[Figure]

**Figure R7.** The dependence of soil $NO_x$ emissions on soil temperature from both flux measurements (gray) and BDISNP scheme (orange). Both observed and BDISNP soil $NO_x$ emissions are binned based on mean soil temperature from WRF-Chem. Three soil temperature bins are described with 4k intervals. The whisker box shows the distribution within each bin and the line connects median soil $NO_x$ emissions across three bins.

Good point. As we use the trimethylbenzene flux to remove the $NO_x$ fluxes impacted by the off-road vehicle emissions (please see our response in Sect. R1.2), we see a much clearer relationshiop between temperature and the RECAP measurements in Figure R7 (also Figure 8 in the context). Statistical analysis is applied and the increase among three bins is statistically significant.

5. It would have been helpful if the authors provided the area for each sector. It would be interesting to know the overall contribution (calculated as area multiplied by flux) of NOx emissions from each sector expressed as tons/day. This would make it easier to understand the source of NOx emissions. We calculated NO2 emissions from the "soil type" land in the SJV using your flux (0.79 mg/m2/h) and the 2017 CropScape. We reached an extraordinarily high soil emission estimate of 747 short tons per day.

We are concerned about scaling the estimated $NO_x$ emissions to SJV or California due to spatial and temporal variation. This study is conducted in June, concurrent with the highest fertilizer application over SJV. Besides, we do not get enough sampling over natural lands, which should have lower soil $NO_x$ emissions. We add the discussion of the limitation of this study in Sect 5.3:

**"It is worth noting the limitation of estimated soil $NO_x$ emissions in our study. First of all, we are unable to investigate the dependence of soil $NO_x$ emissions on meteorological drivers other than soil temperature, such as soil moisture, as modeled soil moisture presents very small variation during the field campaign. Second, as our measurements only cover limited cropland areas in SJV over a short time period and it is around the time of fertilizer use, we cannot scale the estimated soil $NO_x$ emission to the whole year or to the total cropland areas in California. Last, in the absence of ozone and PM2.5 observations, we cannot investigate the impact of soil $NO_x$ emission on air quality. However, as the SJV is in the $NO_x$ limited regime (Pusede et al., 2014), we expect a model that captures the soil $NO_x$ more accurately will produce higher ozone. Future work is needed to further advance our understanding of soil $NO_x$ emission and its role in urban and rural air pollution."**

6. Is there any transport of NOx from highway and urban areas to over croplands?

The transport can be seen from the footprint contour. To avoid the transport from highway and urban areas, we only attribute the observed fluxes with footprints exclusively covering cropland to soil $NO_x$ emission.

7. It is recommended that the authors include a direct comparison of their soil fluxes (0.79 mg NO2/m2/h on average) with those measured in the SJV such as Matson et al. (1997) and Burger and Horwath (2013).

We add the comparison of estimated soil $NO_X$ emission in our study and those from field experiments, including Matson et al. (1997) and Horwath and Burger (2013) in Sect 5.3:

"Soil $NO_x$ emissions in California have been studied in field experiments. Matson et al. (1997) measured soil $NO_x$ emissions from nine dominant crop types in SJV and reported mean fluxes of 0.01-0.09 mg N m$^{-2}$ h$^{-1}$. They also reported a large variation of measured $NO_x$ flux among crops and among different fields of the same crop; the highest measured $NO_x$ flux is 0.17 mg N m$^{-2}$ h$^{-1}$ due to the fertilizer application and soil moisture characteristics. Horwath and Burger (2013) observed an average flux of 0.05-0.28 mg N m$^{-2}$ h$^{-1}$ at mid-days during summertime from five crops in California, and the highest $NO_x$ flux is >4 mg N m$^{-2}$ h$^{-1}$ in systems receiving large N inputs resulting in high concentrations of ammonium. Oikawa et al. (2015) observed soil $NO_x$ emissions in a high-temperature fertilized agricultural region of the Imperial Valley, CA, ranging between -0.02 and 3.2 mg N m$^{-2}$ h$^{-1}$. They also conducted control experiments to investigate the soil $NO_x$ emission responses to fertilization and irrigation. The highest soil $NO_x$ flux was reported ~10 days after the fertilizer at the soil volumetric water content of 30% and the soil temperature of ~313K.

The mean soil $NO_x$ flux, 0.32 mg N m$^{-2}$ h$^{-1}$, derived in our flux measurements is higher than the mean fluxes reported in Matson et al. (1997) and Horwath and Burger (2013), however, the range of estimated soil $NO_x$ flux is within those in Horwath

and Burger (2013) and Oikawa et al. (2015). Fertilizer is likely the primary contributor to the higher mean soil $NO_x$ flux in our study. The RECAP-CA field campaign was conducted in June, right after the month of peak fertilizer use in SJV (Guo et al., 2020). Shown in Oikawa et al. (2015), soil $NO_x$ flux can increase up to 5-fold within 20 days of fertilizer. The higher mean soil $NO_x$ flux is also contributed by higher soil temperature. In our study, the mean soil temperature is 299K with a range between 295K and 304K, whereas the observations in Horwath and Burger (2013) and Oikawa et al. (2015) spread over a wider range of soil temperature, 288K-315K. Consistent with our study, the temperature dependence of soil $NO_X$ emission is observed in these field experiments. Horwath and Burger (2013) reported a 2.5-3.5 fold increase in $NO_x$ fluxes with 10-degree increase in soil temperature. Oikawa et al. (2015) showed that the temperature dependence of soil $NO_x$ emission is non-linear; a steeper increase in soil $NO_x$ emission was observed with the soil temperature exceeding 295K. ”

8. It is important to emphasize that the method they use cannot differentiate between emissions from the soil and emissions from engines on farmland. Therefore, the emissions from croplands should be considered an upper limit. In light of this, the some of the terminology used in the study need to be revised, such as in Figure 4, where the term "soil" need to be changed to a term that reflects this point.

We acknowledge the interference of off-road vehicle emissions over croplands. Even though $NO_x$ flux alone cannot differentiate between emissions from the soil and emissions from off-road vehicle emissions, we decide to use the trimethylbenzene flux to identify the $NO_x$ fluxes impacted by the off-road vehicle emission and remove them in further analysis. Please refer to our response in Sect. R1.2.

9. It is also important to make it clear that the measurements were taken in June, which is near the time when fertilizer is most commonly applied. As a result, the calculated cropland NOx flux is likely the highest it would be throughout the year.

We totally agree and add it in Sect 5.3:

“The mean soil $NO_x$ flux, 0.32 mg N m$^{-2}$ h$^{-1}$, derived in our flux measurements is higher than the mean fluxes reported in Matson et al. (1997) and Horwath and Burger (2013), however, the range of estimated soil $NO_x$ flux is within those in Horwath and Burger (2013) and Oikawa et al. (2015). Fertilizer is likely the primary contributor to the higher mean soil $NO_x$ flux in our study. The RECAP-CA field campaign was conducted in June, right after the month of peak fertilizer use in SJV (Guo et al., 2020). Shown in Oikawa et al. (2015), soil $NO_x$ flux can increase up to 5-fold within 20 days of fertilizer. The higher mean soil $NO_x$ flux is also contributed by higher soil temperature. In our study, the mean soil temperature is 299K with a range between 295K and 304K, whereas the observations in Horwath and Burger (2013) and Oikawa et al. (2015) spread over a wider range of soil temperature, 288K-315K. Consistent with our study, the temperature dependence of soil $NO_X$ emission is observed in these field experiments. Horwath and Burger (2013) reported a 2.5-3.5 fold increase in $NO_x$ fluxes with 10-degree increase in soil temperature. Oikawa et al. (2015) showed that the temperature dependence of soil $NO_x$ emission is non-linear; a steeper increase in soil $NO_x$ emission was observed with the soil temperature exceeding 295K. ”

10. It would be beneficial to have more discussion on why there are differences between the observations and the inventories. Understanding the reasons for these differences would give more insight on how to separate soil NOx from urban/road NOx, and how it could impact ozone and PM2.5 levels in the San Joaquin Valley.

We add the discussion about why the estimated soil $NO_x$ emission is higher in our study in Sect 5.3, please refer to our response to Comment 7 and 9.

11. The error bars in the observed flux (Fig. 5) are considerably large, whereas the error bars in Fig. 4 are comparatively small, particularly for soil emissions. There appears to be an inconsistency between these two calculations.

Figure 4 is the component $NO_x$ flux from different land cover type using the the multi-linear regression, the uncertainty is reduced by the number of data points (4391 data points in this case). Figure 5 is the distribution of gridded $NO_x$ emissions at 4km x 4km, the error bar represents the spatial variation of $NO_x$ emission.

**References**

[revised manuscript text omitted]

---

## Referee Report (RR1)

**Second Review of "Direct observations of NOx emissions over the San Joaquin Valley using airborne flux measurements during RECAP-CA 2021 field campaign, Zhu et al., ACP (2023)**

The authors have adequately addressed the major concerns of the two primary referees. The have also acknowledged the concerns of the unsolicited comments and addressed them to the extent possible with the available data. This paper is sufficient for publication after consideration of the following minor revisions. Below, line numbers refer to the author response document, not the manuscript.

L40: Is there any correlation of soil NOx fluxes with TMB flux, especially at the high NOx tail? If not, worth stating in paper.

L52: Won't filtering out fluxes below LOD introduce a high bias in any resulting averages? If so, maybe better to keep them in.

L81 and Fig. R2: Are these uncertainties in 500m-average fluxes? Please clarify.

L161: Are these details regarding BL depth described in the text or SI? Fig. R3 would be good to include in the SI.

L165: It is still not clear to me why you are normalizing by the standard deviation (which is a scalar for each leg). You'd then just have to multiply it back in to get the right units in the flux. This operation isn't detrending, it is normalization and is not, to my knowledge, standard practice for wavelet fluxes. So, why do it? If you have a good reason, please justify in the text.

L464: "Reviewer 3" opens with hostility. The authors handled this rather gracefully.

L507: This discrepancy between what "controls" the slope of the divergence is, I think, an open question (or at least a confusing one). Paradoxically, the reviewer and the authors may both be correct. An analysis of the budget equation at any point in the atmosphere shows that the flux is related to chemistry, advection, and storage:

$$dC(z)/dt = \text{chemistry} + \text{advection} + dF(z)/dz$$

$$dF(z)/dz = \text{chemistry} + \text{advection} - dC(z)/dt$$

integrating from $z = 0$ to $z = zm$ (measurement height) gives

$$F(zm) = (\text{integral of chemical production/loss}) + (\text{integral of advection}) + (\text{storage}) + F(z = 0)$$

It is also true that the flux at the top of the PBL is "fixed" by entrainment, which is defined as the product of the concentration gradient and the entrainment velocity ($w_e$).

$$F(z = zi) = (C(z<zi) - C(z>zi))*w_e$$

If $w_e$ is purely determined by micrometeorology, the system would seem to be over-determined. Perhaps the solution is that $C(z<zi)$ is inherently a function of the terms in the budget equation – I

honestly am not sure of how to resolve this. Regardless, *this paper is not the appropriate place to resolve it*, and my opinion is that the authors have made a strong effort to constrain divergence as well as the data allows while also being honest about the uncertainties.

L518: Are NOx mixing ratios in the free troposphere higher or lower than in the boundary layer? It should be possible to put some constraint on the entrainment flux in this way (at least the sign).

L537: It is also worth mentioning that Sha (2021) is entirely model-based using a parameterization derived from Oikawa. Not sure it is fair to present it as an independent estimate on par with actual measurements.

L540: It is worth stating in the text why you cannot compare directly to Trousdell and citing the paper. This will help address Reviewer 3's concern about the lack of citation of considerable prior work.

L585: So, is there a way to estimate friction velocity from measurements in the mixed layer? If not, do you have any way to validate the HRRR friction velocity, or does it not strongly impact your results? This is something the airborne flux community needs to consider carefully with respect to accurate footprint estimates. And if it's something that needs more work, that'd be worth mentioning in conclusions.

L666: "Statistical analysis is applied and the increase among three bins is statistically significant." Please quantify this in your response and in the revised text.

---

## Author Response (AR2)

**Response to Reviewer**
**"Direct observations of NO$_x$ emissions over the San Joaquin Valley using airborne flux measurements during RECAP-CA 2021 field campaign"**

Qindan Zhu[1,*], Bryan Place[2,**], Eva Y. Pfannerstill[3], Sha Tong[4,5], Huanxin Zhang[4], Jun Wang[4], Clara M. Nussbaumer[1,6], Paul Wooldridge[2], Benjamin C. Schulze[7], Caleb Arata[8], Anthony Bucholtz[9], John H. Seinfeld[7], Allen H. Goldstein[3,8], and Ronald C. Cohen[1,2]

[1]Department of Earth and Planetary Sciences, University of California, Berkeley, Berkeley, CA 94720, United States
[*]Now at Department of Earth, Atmospheric and Planetary Sciences, Massachusetts Institute of Technology, Cambridge, MA, United States of America
[2]Department of Chemistry, University of California, Berkeley, Berkeley, CA 94720, United States
[**]Now at Office of Research and Development, U.S. Environmental Protection Agency, Research Triangle Park, North Carolina 27711, United States
[3]Department of Environmental Science, Policy, and Management, University of California, Berkeley, Berkeley, CA 94720, United States
[4]Department of Chemical and Biochemical Engineering, Center for Global and Regional Environmental Research, and Iowa Technology Institute, University of Iowa, Iowa City, Iowa 52242, United States
[5]Collaborative Innovation Center on Forecast and Evaluation of Meteorological Disasters (CIC-FEMD), Key Laboratory for Aerosol-Cloud-Precipitation of China Meteorological Administration, Nanjing University of Information Science Technology, Nanjing 210044, People's Republic of China
[6]Department of Atmospheric Chemistry, Max Planck Institute for Chemistry, Mainz 55128, Germany
[7]Department of Environmental Science and Engineering, California Institute of Technology, Pasadena, CA 91125, United States
[8]Department of Civil and Environmental Engineering, University of California, Berkeley, Berkeley, CA 94720, United States
[9]Department of Meteorology, Naval Postgraduate School, Monterey, CA 93943, United States

**Correspondence:** Qindan Zhu (qindan_zhu@berkeley.edu) and Ronald C. Cohen (rccohen@berkeley.edu)

We thank the editor for very careful reading of the manuscript. Below we respond to comments. The reviewer's comments will be shown in red, our response in blue, and changes made to the paper are shown in black block quotes. Unless otherwise indicated, page and line numbers correspond to the original paper. Sections, figures, tables, or equations referenced as "R$n$" are numbered within this response; Figures, tables, and equations numbered normally refer to the numbers in the original discussion paper.

The authors have adequately addressed the major concerns of the two primary referees. The have also acknowledged the concerns of the unsolicited comments and addressed them to the extent possible with the available data. This paper is sufficient for publication after consideration of the following minor revisions. Below, line numbers refer to the author response document, not the manuscript.

[Figure]

**Figure R1.** The scatterplot of $NO_x$ flux and trimethylbenzene fluxes with footprints exclusively covering soil land cover type.

10    Thanks so much for the recognition of our response.

L40: Is there any correlation of soil NOx fluxes with TMB flux, especially at the high NOx tail? If not, worth stating in paper.

We do not find any correlation between soil NOx fluxes with TMB flux (Shown in Figure R1). We add it to Sect 3.4:
"**No correlation between trimethylbeneze flux and $NO_x$ flux is found over croplands.** Among all observations ..."

15    L52: Won't filtering out fluxes below LOD introduce a high bias in any resulting averages? If so, maybe better to keep them in.

We do not filter all fluxes below LOD. Instead, we scan through the segments and only filter out fluxes in the segments which have high LODs and hence all fluxes are below the LOD. To address this comment, we calculate the average fluxes with and without this filtering. It turns out that the filtering does not affect the resulting average.

20    L81 and Fig. R2: Are these uncertainties in 500m-average fluxes? Please clarify.

Yes. We clarify it in the last paragraph of Sect 3.6:

"We propagate the total uncertainty from each component using Eqn. 16 and the distribution of total uncertainty of **500m average NO$_x$ flux** is shown in Figure 3 (b)."

L161: Are these details regarding BL depth described in the text or SI? Fig. R3 would be good to include in the SI.

Yes, we have included the description of BL depth in Sect 3.1 and add Fig. R3 in the supplementary (Fig. S2):

"While most of the measurements are within the planetary boundary layer (PBL), the airplane arose above the boundary layer occasionally and these observations above PBL are removed in later analysis. **The PBL heights are determined using the sharp gradient in the dew point, water concentration, toluene concentration and temperature at the soundings conducted during the voyage, and we interpolate the PBL heights to the full duration of the flight. The PBL heights agree well against the hourly PBL heights from the High-Resolution Rapid Refresh (HRRR) product (Figure. S2).** "

L165: It is still not clear to me why you are normalizing by the standard deviation (which is a scalar for each leg). You'd then just have to multiply it back in to get the right units in the flux. This operation isn't detrending, it is normalization and is not, to my knowledge, standard practice for wavelet fluxes. So, why do it? If you have a good reason, please justify in the text.

Sorry for the confusion. After reviewing the wavelet decomposition algorithm, we agree with the reviewer that this operation is normalization, not detrending. Moreover, normalization is not the requisite step for wavelet decomposition. The normalization is only used to produce Figure 1(a). We changed the word "detrend" to "normalize" in the context:

"For two simultaneous time series of NO$_x$ ($W_c(a,b)$) and vertical wind speed ($W_w(a,b)$), we  obtain the wavelet cross-spectrum following Eqn. 1. The Morlet wavelet-specific reconstruction factor $C_\delta$ is 0.776. We then sum up over the full frequency scales to yield a time series of flux (Eqn. 2).

$$E_{c,w}(j) = \frac{\delta t}{C_\delta} \frac{1}{N} \sum_{n=0}^{N-1} [W_c(a,b) \cdot W_w^*(a,b)] \tag{1}$$

$$F(t) = \overline{c'w'} = \frac{\delta t}{C_\delta} \frac{\delta j}{N} \sum_{n=0}^{N-1} \sum_{j=0}^{J} \frac{[W_c(a,b) \cdot W_w^*(a,b)]}{a(j)} \tag{2}$$

Figure 1 exhibits an example of CWT flux calculation. Figure 1 (a) shows the  **normalized** NO$_x$ and vertical wind speed in a straight segment of $\sim 50$ km. The  **normalization** is realized by subtracting out the average followed by dividing the standard deviation of a scalar time series..."

L464: "Reviewer 3" opens with hostility. The authors handled this rather gracefully.

Thanks. We endeavor to address the reviewer's concerns and improve our manuscript.

L507: This discrepancy between what "controls" the slope of the divergence is, I think, an open question (or at least a confusing one). Paradoxically, the reviewer and the authors may both be correct. An analysis of the budget equation at any point in the atmosphere shows that the flux is related to chemistry, advection, and storage: dC(z)/dt = chemistry + advection + dF(z)/dz dF(z)/dz = chemistry + advection – dC(z)/dt integrating from z = 0 to z = zm (measurement height) gives F(zm) = (integral of chemical production/loss) + (integral of advection) + (storage) + F(z = 0) It is also true that the flux at the top of the PBL is "fixed" by entrainment, which is defined as the product of the concentration gradient and the entrainment velocity $(w_e).F(z = zi) = (C(z < zi)\breve{}C(z > zi)) * w$e If we is purely determined by micrometeorology, the system would seem to be over-determined. Perhaps the solution is that C(z<zi) is inherently a function of the terms in the budget equation – I honestly am not sure of how to resolve this. Regardless, this paper is not the appropriate place to resolve it, and my opinion is that the authors have made a strong effort to constrain divergence as well as the data allows while also being honest about the uncertainties.

Thanks so much for this comment. We actually agree with the reviewer that the treatment of vertical divergence is quite confusing in current studies related to flux analysis. That is the main reason for us to conservatively account for the uncertainty from the vertical divergence.

L518: Are NOx mixing ratios in the free troposphere higher or lower than in the boundary layer? It should be possible to put some constraint on the entrainment flux in this way (at least the sign).

NOx mixing ratios are lower in the free troposphere than in the boundary layer, so the entrainment flux is negative. It is consistent with our calculation (-0.09 mg N m$^{-2}$ h$^{-1}$) at the boundary layer top. However, we have to emphasize that the uncertainty in the calculation can be large.

L537: It is also worth mentioning that Sha (2021) is entirely model-based using a parameterization derived from Oikawa. Not sure it is fair to present it as an independent estimate on par with actual measurements.

Since Guo et al. (2020) and Almaraz et al. (2018) are also model-based, we think it is fair to present Sha et al. (2021) on par with Guo et al. (2020) and Almaraz et al. (2018).

L540: It is worth stating in the text why you cannot compare directly to Trousdell and citing the paper. This will help address Reviewer 3's concern about the lack of citation of considerable prior work.

It is a good point. In Sect 5.3 describing the limitation of our study, we add:
"Second, as our measurements only cover limited cropland areas in SJV over a short time period and it is around the time of fertilizer use, we cannot scale the estimated soil NOx emission to the whole year or to the total cropland areas in California. **Therefore, we cannot directly compare our estimate of soil NO$_x$ emission against other studies reporting soil NO$_x$ emissions on an annual basis or on a larger spatial scale.**"

L585: So, is there a way to estimate friction velocity from measurements in the mixed layer? If not, do you have any way to validate the HRRR friction velocity, or does it not strongly impact your results? This is something the airborne flux community needs to consider carefully with respect to accurate footprint estimates. And if it's something that needs more work, that'd be worth mentioning in conclusions.

Friction velocity accounts for shear stress in the turbulent boundary layer and it can be parameterized from measurements. For instance, Foken and Napo (2008) and Amiro (1990) presented the calculation of friction velocity using the instantaneous wind vectors ($u'$, $v'$, $w'$). Hannun et al. (2020) calculated the friction velocity using the mean flux parameterization. Camuffo (2014) showed four different parameterizations of friction velocity under different turbulence conditions. We use the friction velocity from HRRR since the roughness layer, another parameter used in the footprint calculation, is also from HRRR. We also test out the effect of friction velocity on footprint by varying the friction velocity by 50%, and it does not affect the result.

L666: "Statistical analysis is applied and the increase among three bins is statistically significant." Please quantify this in your response and in the revised text.

We quantify the temperature dependence of soil $NO_x$ emission using the slope from a linear fit between soil temperature and estimated soil $NO_x$ emissions. We add it in Sect 5.2

"A range of soil temperature between 295K to 304K is observed. **We apply the linear regression fit between soil temperature and estimated soil $NO_x$ emissions and show a positive temperature dependence of soil $NO_x$ emissions with the slope of 0.02 ($\pm$0.008) mg N m$^{-2}$ h$^{-1}$ K$^{-1}$.** We then bin observed soil $NO_x$ emissions to three soil temperature categories...
"

**References**

Almaraz, M., Bai, E., Wang, C., Trousdell, J., Conley, S., Faloona, I., and Houlton, B. Z.: Agriculture is a major source of NO x pollution in California, Science advances, 4, eaao3477, 2018.

Amiro, B.: Comparison of turbulence statistics within three boreal forest canopies, Boundary-Layer Meteorology, 51, 99–121, 1990.

100  Camuffo, D.: Atmospheric stability and pollutant dispersion, Microclimate for Cultural Heritage, pp. 245–282, 2014.

Foken, T. and Napo, C. J.: Micrometeorology, vol. 2, Springer, 2008.

Guo, L., Chen, J., Luo, D., Liu, S., Lee, H. J., Motallebi, N., Fong, A., Deng, J., Rasool, Q. Z., Avise, J. C., et al.: Assessment of nitrogen oxide emissions and San Joaquin Valley PM2. 5 impacts from soils in California, Journal of Geophysical Research: Atmospheres, 125, e2020JD033 304, 2020.

105  Hannun, R. A., Wolfe, G. M., Kawa, S. R., Hanisco, T. F., Newman, P. A., Alfieri, J. G., Barrick, J., Clark, K. L., DiGangi, J. P., Diskin, G. S., et al.: Spatial heterogeneity in CO2, CH4, and energy fluxes: Insights from airborne eddy covariance measurements over the Mid-Atlantic region, Environmental Research Letters, 15, 035 008, 2020.

Sha, T., Ma, X., Zhang, H., Janechek, N., Wang, Y., Wang, Y., Castro Garci´a, L., Jenerette, G. D., and Wang, J.: Impacts of Soil NO x Emission on O3 Air Quality in Rural California, Environmental science & technology, 55, 7113–7122, 2021.